# NeSyPr: Neurosymbolic Proceduralization For Efficient Embodied Reasoning

**Wonje Choi, Jooyoung Kim, Honguk Woo***
Department of Computer Science and Engineering, Sungkyunkwan University
{wjchoi1995, onsaemiro, hwoo}@skku.edu

## Abstract

We address the challenge of adopting language models (LMs) for embodied tasks in dynamic environments, where online access to large-scale inference engines or symbolic planners is constrained due to latency, connectivity, and resource limitations. To this end, we present NESYPR, a novel embodied reasoning framework that compiles knowledge via neurosymbolic proceduralization, thereby equipping LM-based agents with structured, adaptive, and timely reasoning capabilities. In NESYPR, task-specific plans are first explicitly generated by a symbolic tool leveraging its declarative knowledge. These plans are then transformed into composable procedural representations that encode the plans' implicit production rules, enabling the resulting composed procedures to be seamlessly integrated into the LM's inference process. This neurosymbolic proceduralization abstracts and generalizes multi-step symbolic structured path-finding and reasoning into single-step LM inference, akin to human knowledge compilation. It supports efficient test-time inference without relying on external symbolic guidance, making it well suited for deployment in latency-sensitive and resource-constrained physical systems. We evaluate NESYPR on the embodied benchmarks PDDLGym, VirtualHome, and ALFWorld, demonstrating its efficient reasoning capabilities over large-scale reasoning models and a symbolic planner, while using more compact LMs.

## 1 Introduction

Recent works such as Inner Monologue [1], SayCan [2], and LLM-Planner [3] have demonstrated the potential of large-scale language models (LMs) to control embodied agents on complex tasks in dynamic environments. Yet, the inherent limitations of autoregressive inference (e.g., shallow planning, inefficient context reuse, and lack of structure) have led researchers to explore more structured reasoning approaches. Three primary directions have emerged: (i) agentic frameworks that support autonomous planning and multi-step reasoning [4, 5, 6, 7], (ii) neurosymbolic methods that integrate LMs with external symbolic reasoning frameworks [8, 9, 10, 11], and (iii) augmented LM approaches that incorporate memory modules [12, 13, 14, 15]. Despite these advances, existing approaches still face significant limitations, particularly in resource-constrained dynamic environments. Agentic frameworks require iterative inference of large-scale models, leading to substantial computational overhead [16]. Neurosymbolic methods, while using predefined rules to systematically search for accurate reasoning paths, often suffer from increased task-solving time and diminished effectiveness in dynamic settings unless the solver and its rule base are continuously updated to reflect environmental changes [17]. Memory-augmented LM approaches typically expand the context window to encode more information, yet they rarely retain the procedural structure needed for complex embodied tasks.

These limitations emphasize the need for a reasoning framework tailored to LM-based embodied agents, capable of structured and adaptive decision-making under time and resource constraints and

---

*Honguk Woo is the corresponding author.

39th Conference on Neural Information Processing Systems (NeurIPS 2025).

without online symbolic assistance. To this end, we draw inspiration from the Adaptive Control of Thought (ACT) theory [18], which models skill acquisition as a process of knowledge compilation—the conversion of declarative knowledge into procedural form via repeated practice. Declarative knowledge is held in declarative memory as chunks that encode explicit facts such as propositions or problem states. By contrast, procedural knowledge consists of condition–action rules, stored in procedural memory and triggered automatically as cognitive procedures. A central mechanism in ACT, known as *proceduralization*, hinges on the interplay of three memory systems. Declarative memory supplies the relevant chunk into working memory, which temporarily holds the current problem state during interaction with the environment. Through repeated use, the factual patterns are gradually compiled into production rules stored in procedural memory. Once compiled, production rules fire whenever their conditions match the contents of working memory, enabling actions without further reference to declarative memory. By bypassing declarative retrieval, proceduralization reduces cognitive load and supports faster, more automatic, and less error-prone execution [19, 20, 21, 22].

Building on the ACT theory, we present NESYPR, a novel embodied reasoning framework based on neurosymbolic proceduralization. It performs knowledge compilation by abstracting and generalizing multi-step symbolic path-finding and reasoning into a single-step LM inference, thereby enabling LM-based agents to perform embodied reasoning efficiently, without relying on large-scale inference or online access to external symbolic tools. In NESYPR, task-specific plans are first explicitly generated by a symbolic tool leveraging its declarative knowledge. These plans are then transformed into composable procedural representations that encode the plans' implicit production rules, enabling the resulting composed procedures to be seamlessly integrated into the LM's inference process. This proceduralization proceeds in two phases: i) compositional NeSy procedure learning and ii) NeSy procedure contrastive planning. During training phase i), an agent learns to encode production rules into a vector-quantized procedural memory, in which the resulting vectors are structured to be composable for task-specific plan generation. At test phase ii), without access to symbolic tools, the agent continues adaptive reasoning by generating plans augmented with procedural memory, while contrastively reconstructing their internal representations based on environmental feedback.

We evaluate NESYPR on PDDLGym [23], VirtualHome [24], and ALFWorld [25], where inputs include observation, goal, and domain knowledge that are specified symbolically. For structured reasoning, NESYPR achieves a 46.7% higher task success rate than **DeepSeek-R1-Distill** [26], a distilled 70B-scale reasoning model, while operating with a 70 times smaller LM (as shown in Table 3). For adaptive reasoning, it attains a 62.1% higher success rate on unseen tasks with dynamic conditions than the **symbolic planner** [27] (in Table 3). For timely reasoning, it reduces inference latency by more than 90.0% compared to **BoT** [28], a large-scale inference baseline, while achieving a 36.0% improvement in task success rate (in Table 3). These results demonstrate that NESYPR endows LM-based agents with strong structured, adaptive, and timely reasoning capabilities.

Our contributions are summarized as: (1) We present NESYPR, the neurosymbolic proceduralization-based reasoning framework inspired by ACT, which compiles multi-step symbolic reasoning into single-step LM inference, eliminating the need for online symbolic planners in embodied tasks. (2) We develop compositional NeSy procedure learning, which encodes production rules into a vector-quantized procedural memory whose vectors can be compositionally combined to generate task-specific plans. (3) We implement NeSy procedure contrastive planning, which adaptively generates plans by contrastively reconstructing task-specific procedures from stored procedures labeled with environmental feedback. (4) We show the effectiveness and efficiency of NESYPR through extensive evaluations including 3 embodied benchmarks and 9 experimental scenarios, demonstrating its capabilities for structured, adaptive, and timely reasoning.

## 2   Related Work

**Agentic Frameworks for Embodied Tasks.** A growing body of research has explored how LMs can be utilized to plan actions in physical or simulated environments [1, 2, 3, 29, 30, 31, 32]. Recent studies emphasize agentic frameworks that enable autonomous planning and multi-step reasoning, rather than relying on single-step predictions. Within this paradigm, methods such as ReAct [4], Reflexion [5], and others [6, 33, 34, 7, 35, 36] integrate Chain-of-Thought (CoT) [37, 38, 39, 40] reasoning with environmental feedback. Although these frameworks demonstrate strong performance, they typically rely on iterative inference of large-scale models, leading to substantial computational

overhead. In contrast, our approach employs an LM equipped with a specialized memory architecture, enabling robust reasoning without dependence on large-scale models or multi-step inference.

**Neurosymbolic Methods for Embodied Task Reasoning.** Neurosymbolic approaches integrate symbolic reasoning modules—such as rule-based or logic programming tools [41, 42, 43]—with neural networks to achieve interpretable and verifiable reasoning. With the advent of LMs, these hybrid systems have advanced logical reasoning in natural language tasks [44, 45, 46, 47, 48]. This line of research has also extended into embodied tasks [8, 9, 10, 11, 49], where delegating task reasoning to symbolic tools yielded more reliable and optimized action plans. However, these methods depend on handcrafted domain knowledge (e.g., action rules) and require continuous updates to remain effective in dynamic environments [17]. Moreover, solving time increases sharply with task complexity, limiting real-time decision-making in complex settings. In contrast, our approach embeds procedural knowledge within the LM's memory architecture, enabling efficient reasoning and end-to-end adaptation through environmental feedback, without online symbolic assistance.

**Memory-augmented LMs for Long-term Generation.** Recent studies [14, 15, 50, 51, 52, 53] enhance LMs with external memory structures to better retain long-term context, often through recurrent memory updates within transformer architectures. Other approaches store intermediate attention states—such as key-value pairs from relevant documents or histories—in external memory modules for retrieval during inference [12, 13, 54]. While these methods expand the model's context window to capture richer semantic information, only a few studies [55, 56, 57] explore memory architectures specialized for embodied tasks. In contrast, our approach introduces a procedural memory architecture that encodes task-level procedural knowledge, enabling efficient reasoning and adaptive behavior in dynamic embodied environments.

## 3 Problem formulation

We consider embodied reasoning in dynamic settings, where an agent engages with a stream of tasks and must adapt to changing states and goals over time. Each task is defined as a tuple $\tau = (\mathcal{S}, \mathcal{A}, \mathcal{P}, g)$, where $s \in \mathcal{S}$ is the state, $a \in \mathcal{A}$ is the action, $\mathcal{P} : \mathcal{S} \times \mathcal{A} \to \mathcal{S}$ is the transition function describing dynamics, $g \in \mathcal{G}$ denotes the goal. Due to partial observability [58], the agent receives an observation $o_t$ at each timestep $t$. Unlike conventional multitask settings [1, 3], the agent must solve a sequence of tasks $\mathcal{T} = \{\tau_1, \tau_2, \ldots, \tau_N\}$, over time, where both $g$ and $\mathcal{P}$ may vary across tasks [59, 60]. Our objective is to develop an LM-based agent (LM policy) that solves tasks autonomously and continuously at test time with no online access to any symbolic tools, while internalizing procedural knowledge from symbolic guidance during training. Note that Eq. (1) defines the ideal objective, which is approximated in practice by supervising the LM on planner-computed action sequences with symbolic inputs.

$$\pi_{\text{LM}}^* = \underset{\pi_{\text{LM}}}{\arg\max} \sum_{i=1}^{N} \mathbb{E}_{\tau_i} \left[ \sum_{t=0}^{T} \text{SR}(s_t, \pi_{\text{LM}}(o_t, g)) - \text{D}_{\text{KL}}(\pi_{\text{LM}}(\cdot \mid o_t, g) \parallel \pi_{\text{tool}}(\cdot \mid o_t, g)) \right] \quad (1)$$

Here, $\text{SR} : \mathcal{S} \times \mathcal{A} \to \{0, 1\}$ indicates whether actions taken in current states $s_t$ lead to task success, and $\text{D}_{\text{KL}}$ measures the divergence between the LM-based policy $\pi_{\text{LM}}$ and the symbolic policy $\pi_{\text{tool}}$ derived from external tools in [42, 27, 61]. Accordingly, $\pi_{\text{LM}}^*$ aims at maximizing task success while aligning its learned procedural behavior with the tool's declarative guidance.

## 4 NᴇSʏPʀ: Neurosymbolic Proceduralization

To equip agents with structured and adaptive reasoning for diverse embodied tasks, NᴇSʏPʀ learns to encode production rules and compose procedures, compressed representations derived from the declarative knowledge of symbolic tools. We refer to this end-to-end learning and utilization process as *neurosymbolic proceduralization*. As illustrated in Figure 1, neurosymbolic proceduralization operates in two phases: i) a training phase, *compositional NeSy procedure learning*, where procedural knowledge is structured within procedural memory using plans generated by a symbolic tool, and ii) a test phase, *NeSy procedure contrastive planning*, where the agent autonomously adapts to new tasks by contrastively reconstructing procedural representations, without relying on symbolic tools.

During phase i), the agent trains on offline data comprising symbolically defined problem instances (observations and goals) and associated domain knowledge (action rules). The declarative knowledge

Figure 1: The framework architecture of NESYPR

used by symbolic tools for problem-solving, such as search algorithms, state transitions, cost estimation, and goal evaluation, is internalized as production rules. The agent composes these rules into task-solving procedures in procedural memory, which it then exploits to generate plans. In phase ii), using the procedural memory established during phase i) training, the agent performs structured reasoning without access to external symbolic tools (i.e., declarative memory). It further engages in adaptive reasoning by contrastively reconstructing procedures from prior ones labeled as successes or failures via environmental feedback. The agent continually reinforces plans aligned with valid procedures while suppressing those associated with invalid ones. Accordingly, NESYPR enables LM-based agents to reason robustly across tasks and adapt efficiently to ever-changing environments.

## 4.1 Compositional NeSy Procedure Learning

As shown in Figure 2, our learning method incorporates a procedural memory that extends existing approaches [15, 14] across $L$ layers to support structured reasoning. Working memory $M$ extends the context window by accumulating symbolic inputs from the environment. The procedural memory then performs vector quantization (VQ) [62], encoding production rules into discrete procedure-units stored in a procedure-book $\mathcal{C}$, which are composed to generate plans.

$$\boldsymbol{H}_l, \boldsymbol{M} = \text{DecoderBlock}_l\big(\boldsymbol{H}_{l-1}, \boldsymbol{M}\big), \quad \boldsymbol{M} \triangleq [\boldsymbol{e}_1, \boldsymbol{e}_2, \ldots, \boldsymbol{e}_S], \; \boldsymbol{e}_i \in \mathbb{R}^D \tag{2}$$

At each layer $l \in \{1, \ldots, L\}$, the decoder block $\text{DecoderBlock}_l$ takes the previous hidden state $\boldsymbol{H}_{l-1}$ and $\boldsymbol{M}$ as input. Each slot $\boldsymbol{e}_i$ encodes environmental context in dimension $D$, with $S$ defining memory capacity. Runtime procedure $\boldsymbol{R}$ integrated into $\text{DecoderBlock}_l$ contributes to refining $\boldsymbol{H}_l$.

**Memory-augmented module.** $\boldsymbol{M}$ encodes the current environmental state, using a memory-augmented cross-attention adapted from [15]. To enable information exchange between the self-attention output $\boldsymbol{E}_{\text{self}} \in \mathbb{R}^{T \times D}$ from $\boldsymbol{H}_{l-1}$ and $\boldsymbol{M}$, we apply a cross-attention.

$$\boldsymbol{E}_{\text{work}} = \text{softmax}\left(\frac{QK^\top}{\sqrt{D}}\right)V, \quad Q = \boldsymbol{E}_{\text{self}}\boldsymbol{W}_Q, K = \boldsymbol{M}\boldsymbol{W}_K, V = \boldsymbol{M}\boldsymbol{W}_V \tag{3}$$

Here, $\boldsymbol{W}_Q, \boldsymbol{W}_K, \boldsymbol{W}_V \in \mathbb{R}^{D \times D}$ are learnable projection matrices. $\boldsymbol{M}$ is then updated via a gating mechanism that merges the original memory with the cross-attended representation $\boldsymbol{E}_{\text{work}}$.

$$\boldsymbol{M} \leftarrow g_{\text{up}} \odot \alpha(\boldsymbol{E}_{\text{work}}) + (1 - g_{\text{up}}) \odot \boldsymbol{M}, \quad g_{\text{up}} = \sigma\left(\alpha(\boldsymbol{E}_{\text{work}})\boldsymbol{W}_{\text{up}}\right) \tag{4}$$

Here, $\alpha$ is alignment operator ensuring dimensional consistency, $\sigma$ denotes the sigmoid activation function, $\odot$ represents slot-wise multiplication, and $\boldsymbol{W}_{\text{up}} \in \mathbb{R}^{D \times D}$ is a learnable projection matrix.

In *procedural memory*, $\boldsymbol{R}$ is obtained by applying VQ to $\boldsymbol{M}$ using a procedure-book $\mathcal{C} = \{c_1, c_2, \ldots, c_K\}$, where each procedure-unit $c_j \in \mathbb{R}^d$ is a $d$-dimensional vector. Each slot $\boldsymbol{e}_i \in \boldsymbol{M}$ is partitioned into contiguous $d$-dimensional chunks to align with the procedure-units.

$$\boldsymbol{e}_i = \big[e_i^{(1)}; e_i^{(2)}; \ldots; e_i^{(q)}\big], \quad q = \lfloor D/d \rfloor, \; e_i^{(r)} \in \mathbb{R}^d \tag{5}$$

Each $e_i^{(r)}$ is replaced with its nearest procedure-units $c_{k_r} \in \mathcal{C}$, selected by minimizing the Euclidean distance.

$$\boldsymbol{c}_i = \big[c_{k_1}; c_{k_2}; \ldots; c_{k_q}\big], \quad c_{k_r} = \underset{c_j \in \mathcal{C}}{\arg\min} \|e_i^{(r)} - c_j\|_2 \tag{6}$$

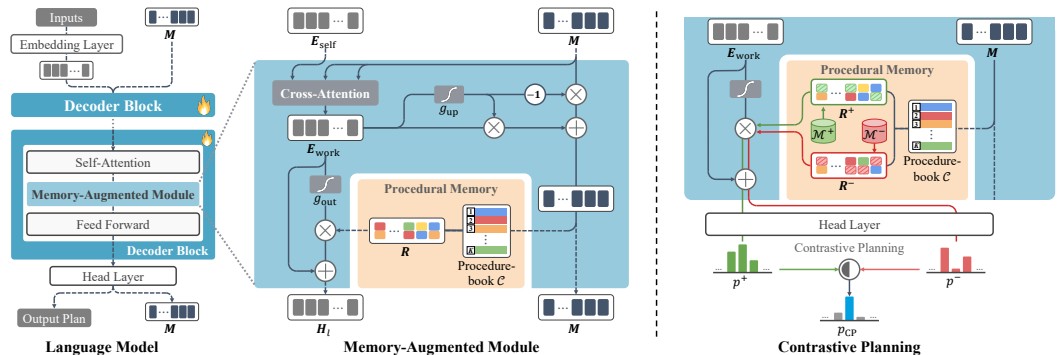

Figure 2: Vector-quantized procedural memory and contrastive planning

Concatenating the selected procedure-units forms a composite procedure $c_i$. Aggregating all such procedures across memory slots yields runtime procedure: $\boldsymbol{R} = [\boldsymbol{c}_1, \boldsymbol{c}_2, \dots, \boldsymbol{c}_S] \in \mathbb{R}^{S \times D}$. To integrate $\boldsymbol{R}$ into the model reasoning process, we combine $\boldsymbol{R}$ with $\boldsymbol{E}_{\text{work}}$, enhancing $\boldsymbol{H}_l$.

$$\boldsymbol{H}_l = \text{FFN}\left(\boldsymbol{E}_{\text{work}} + g_{\text{out}} \odot \alpha(\boldsymbol{R})\right), \quad g_{\text{out}} = \sigma\left(\boldsymbol{E}_{\text{work}}\boldsymbol{W}_{\text{out}}\right), \boldsymbol{W}_{\text{out}} \in \mathbb{R}^{D \times D} \tag{7}$$

Here, FFN is a feed-forward submodule, $\boldsymbol{W}_{\text{out}}$ is a learnable matrix, and $g_{\text{out}}$ is a gating matrix.

**Learning objective.** To train the procedure-book $\mathcal{C}$ end-to-end for compositional procedures, we combine the task objective (e.g., LM fine-tuning) with a VQ loss applied at each layer $l$.

$$\mathcal{L}_{\text{VQ}}^{(l)} = \|\text{sg}(\boldsymbol{M}^{(l)}) - \boldsymbol{R}^{(l)}\|_F^2 + \beta\|\boldsymbol{M}^{(l)} - \text{sg}(\boldsymbol{R}^{(l)})\|_F^2 \tag{8}$$

Here, sg is the stop-gradient operator, $\|\cdot\|_F$ is the Frobenius norm [63], and $\beta$ is a weighting coefficient that controls $\boldsymbol{M}$ to align with $\boldsymbol{R}$. The overall training objective is defined as

$$\mathcal{L} = -\mathbb{E}_{\mathcal{T}}\left[\log \pi_\theta(a \mid o, g, \boldsymbol{M})\right] + \lambda \sum_{l=1}^{L} \mathcal{L}_{\text{VQ}}^{(l)} \tag{9}$$

where $\theta$ denotes the learnable parameters of the LM, and $\lambda$ balances the task-specific objective and the VQ term. $\mathcal{C}$ is updated using an exponential moving average (EMA) [64], providing compositional procedural representations to $\boldsymbol{R}$ for use in structured reasoning.

## 4.2 NeSy Procedure Contrastive Planning

To support adaptive reasoning at test time without symbolic tools, we introduce a procedure-based contrastive planning strategy that reconstructs composed procedures from feedback-labeled stored procedures and contrastively generates plans aligned with dynamic environments.

**Procedure reconstruction.** We maintain two procedure banks: $\mathcal{M}^+$ for successful procedures and $\mathcal{M}^-$ for failures. For each $\boldsymbol{c}_i \in \boldsymbol{R}$, before integration in Eq. (7), we reconstruct two versions: a positive procedure $\boldsymbol{c}_i^+$ by matching against $\mathcal{M}^+$, and a negative procedure $\boldsymbol{c}_i^-$ against $\mathcal{M}^-$.

$$\boldsymbol{c}_i^+ \leftarrow \begin{cases} \boldsymbol{c}^+ & \text{if } \exists \boldsymbol{c}^+ \in \mathcal{M}^+ \wedge \text{sim}(\boldsymbol{c}_i, \boldsymbol{c}^+) \geq \upsilon \\ \boldsymbol{c}_i & \text{otherwise} \end{cases}, \boldsymbol{c}_i^- \leftarrow \begin{cases} \boldsymbol{c}^- & \text{if } \exists \boldsymbol{c}^- \in \mathcal{M}^- \wedge \text{sim}(\boldsymbol{c}_i, \boldsymbol{c}^-) \geq \upsilon \\ \boldsymbol{c}_i & \text{otherwise} \end{cases} \tag{10}$$

Here, sim denotes a similarity function (e.g., cosine similarity), and $\upsilon$ is a reconstruction threshold. We then reconstruct two runtime procedures: $\boldsymbol{R}^+ = [\boldsymbol{c}_1^+, \boldsymbol{c}_2^+, \dots, \boldsymbol{c}_S^+]$ and $\boldsymbol{R}^- = [\boldsymbol{c}_1^-, \boldsymbol{c}_2^-, \dots, \boldsymbol{c}_S^-]$. These are used to generate two versions of the hidden state $\boldsymbol{H}_l$ within a single batch: one conditioned on $\boldsymbol{R}^+$ and the other on $\boldsymbol{R}^-$. Both are then used in contrastive decoding [65].

**Contrastive planning.** Following [66], we guide generation toward successful procedures by computing a contrastive score for each token $x_i$ within an adaptive plausibility set $\mathcal{V}_{\text{head}}(x_{<i})$,

$$\mathcal{V}_{\text{head}}(x_{<i}) = \{x_i \in \mathcal{V} \mid p^+(x_i \mid x_{<i}, \boldsymbol{M}; \mathcal{M}^+) \geq \vartheta \max_{x'} p^+(x' \mid x_{<i}, \boldsymbol{M}; \mathcal{M}^+)\} \tag{11}$$

where $\vartheta = 0.1$ controls the truncation threshold. For $x_i \in \mathcal{V}_{\text{head}}(x_{<i})$, we obtain contrastive score by

$$S(x_i) = \log p^+(x_i \mid x_{<i}, \boldsymbol{M}; \mathcal{M}^+) - \log p^-(x_i \mid x_{<i}, \boldsymbol{M}; \mathcal{M}^-) \tag{12}$$

where $p^+$ and $p^-$ denote token distributions conditioned on successful and failed procedures, respectively. The final next-token distribution is then reshaped as follows.

$$p_{\mathrm{CP}}(x_i \mid x_{<i}) = \begin{cases} \mathrm{softmax}\left(S(x_i)\right) \cdot \sum_{x' \in \mathcal{V}_{\mathrm{head}}} p^+(x' \mid x_{<i}) & \text{if } x_i \in \mathcal{V}_{\mathrm{head}}(x_{<i}) \\ p^+(x_i \mid x_{<i}) & \text{otherwise} \end{cases} \tag{13}$$

This decoding process suppresses failure patterns and promotes plans aligned with the environment, enabling adaptive reasoning without symbolic guidance. Algorithm of NESYPR is in Appendix.

## 5 Evaluation

### 5.1 Experiment Setting

**Environments.** We evaluate NESYPR across diverse embodied benchmarks, including multiple domains from PDDLGym [23] (e.g., Minecraft, Rearrangement, GlibRearrangement), VirtualHome [67], and ALFWorld [25]. To assess embodied reasoning performance in dynamic environments, as described in Section 3, we configure each benchmark to support task sequences that require multi-step planning and continual adaptation. In PDDLGym, where symbolic planners are built-in [27, 61], observations are provided in symbolic form. We construct 9 distinct task sequences by randomly composing tasks, ensuring consistent evaluation settings across all baselines. For VirtualHome and ALFWorld, we utilize symbolic observation interfaces provided by their respective open-source implementation. We evaluate under continual task settings with behavior-incremental and environment-incremental configurations, using 4 distinct task sequences for each, following [68]. During the evaluation, agents receive only binary feedback at the task level (success or failure), and no gradient updates are allowed.

**Datasets.** For training, we use a small set of problem instances paired with plans generated by symbolic planners [27]. In PDDLGym, the train sets include 29 instances for Minecraft, 20 for Rearrangement, and 40 for GlibRearrangement. The test sets contain 389, 400, and 80 instances respectively, all disjoint from the train data. For VirtualHome and ALFWorld, the train sets consist of 77 and 549 instances, respectively. Each test set is split into seen and unseen sets. The seen set contains 112 and 1,509 instances respectively, and shares the same goal as the train set, but varies in object placement and inter-object relations. The unseen set contains 52 and 1,369 instances respectively, and introduces entirely new tasks, not present in both the train and the seen sets. At test time, agents are given only the current observation and goal, with no access to symbolic tools.

**Baselines.** For comparison, we organize the baselines into four categories: (i) Single-step planning, including ZSP [30], RAP [69], and LLM-Planner [3], generates action plans in an inference step. (ii) Agentic workflow, such as CoT [37], ToT [38], GoT [39], ReAct [4], and Reflexion [5], performs multi-step reasoning through multiple LM calls. (iii) Memory-augmented LM, including LongMem [12] and LM2 [15], incorporates long-term memory into the LM attention mechanism. Optimus-2 [55] and DT-Mem [57] extend this approach for embodied agents. We also include BUTLER [70], a parameter-efficient fine-tuning method. (iv) Proceduralization such as BoT [28] stores abstract reasoning templates in a meta-buffer and dynamically instantiates them to guide procedural LM reasoning. Furthermore, Large Reasoning Model (LRM) [71] integrates CoT-style reasoning through reinforcement learning, with compact variants distilled from larger models. PlaSma [72] distills procedural knowledge from larger LMs into compact models. By default, we use LLaMA-3.2-1B [73] for PDDLGym, and Qwen2.5-0.5B [74] for VirtualHome and ALFWorld.

**Metrics.** We use four standard metrics, following [30, 75, 76]. Cumulative Task Success Rate (CSR) measures the percentage of tasks where all sub-goals are achieved. Cumulative Goal-Conditioned Success Rate (CGC) reports the fraction of individual sub-goals achieved across all tasks. Executability (Exe) assesses if each selected action is feasible. Success rate weighted by Path Length (SPL) reflects both task success and path efficiency.

Further details of the experimental settings are provided in the Appendix.

### 5.2 Main Result

**Open-loop continual embodied tasks.** To evaluate the generalization performance of NESYPR on open-loop continual task planning, we conduct experiments in Table 1 using multiple domains from PDDLGym. In this setting, the agent generates a complete action sequence without intermediate

Table 1: Performance on open-loop continual embodied tasks in PDDLGym. Metrics are averaged over 9 random seeds, with standard deviations to indicate consistency across runs. PARAMS denotes the total number of model parameters, with the ratio of trainable parameters in parentheses.

| METHOD | PARAMS | TRAIN | | | | TEST | | | |
|---|---|---|---|---|---|---|---|---|---|
| | | CSR (↑) | CGC (↑) | EXE (↑) | SPL (↑) | CSR (↑) | CGC (↑) | EXE (↑) | SPL (↑) |
| **DOMAIN: MINECRAFT** | | | | | | | | | |
| ZSP | 1.7B (0.0%) | 76.4±5.5 | 81.7±4.8 | 100.0±0.0 | 0.8±0.1 | 16.4±1.5 | 35.9±2.0 | 100.0±0.1 | 0.1±0.0 |
| RAP | 1.7B (0.0%) | 76.4±5.5 | 81.7±4.8 | 100.0±0.0 | 0.8±0.1 | 16.8±0.9 | 18.0±1.9 | 100.0±0.1 | 0.1±0.0 |
| CoT | 1.7B (0.0%) | 83.3±6.7 | 83.3±9.3 | 100.0±0.0 | 0.8±0.1 | 17.0±0.5 | 25.7±1.8 | 100.0±0.0 | 0.1±0.0 |
| ToT | 1.7B (0.0%) | 85.1±4.2 | 85.7±3.7 | 100.0±0.0 | 0.9±0.0 | 18.7±1.0 | 32.2±2.6 | 100.0±0.0 | 0.1±0.0 |
| GoT | 1.7B (0.0%) | 89.1±5.1 | 94.6±7.7 | 100.0±0.0 | 0.9±0.1 | 18.9±0.5 | 25.9±1.2 | 100.0±0.0 | 0.1±0.0 |
| BUTLER | 1.2B (0.6%) | 100.0±0.0 | 100.0±0.0 | 100.0±0.0 | 1.0±0.0 | 51.4±1.9 | 56.7±2.6 | 99.7±0.3 | 0.4±0.0 |
| LONGMEM | 1.6B (24.6%) | 100.0±0.0 | 100.0±0.0 | 100.0±0.0 | 1.0±0.0 | 53.3±3.7 | 56.3±4.1 | 99.8±0.1 | 0.5±0.0 |
| LM2 | 1.3B (6.3%) | 100.0±0.0 | 100.0±0.0 | 100.0±0.0 | 1.0±0.0 | 47.9±8.4 | 56.5±4.6 | 99.9±0.1 | 0.4±0.0 |
| NESYPR | 1.3B (6.3%) | 100.0±0.0 | 100.0±0.0 | 100.0±0.0 | 1.0±0.0 | **65.2**±1.4 | **68.9**±2.4 | 100.0±0.1 | **0.6**±0.0 |
| **DOMAIN: REARRANGEMENT** | | | | | | | | | |
| ZSP | 1.7B (0.0%) | 94.2±3.8 | 97.6±1.2 | 100.0±0.0 | 0.9±0.0 | 15.3±2.1 | 22.4±3.4 | 100.0±0.0 | 0.1±0.0 |
| RAP | 1.7B (0.0%) | 94.2±3.8 | 97.6±1.2 | 100.0±0.0 | 0.9±0.0 | 18.3±1.7 | 29.0±2.6 | 100.0±0.0 | 0.1±0.0 |
| CoT | 1.7B (0.0%) | 95.0±4.5 | 98.5±2.5 | 100.0±0.0 | 1.0±0.0 | 19.2±0.9 | 29.9±1.5 | 100.0±0.0 | 0.1±0.0 |
| ToT | 1.7B (0.0%) | 95.0±6.3 | 99.5±1.2 | 100.0±0.0 | 1.0±0.1 | 20.2±1.2 | 33.9±1.4 | 100.0±0.0 | 0.2±0.0 |
| GoT | 1.7B (0.0%) | 95.8±4.9 | 100.0±0.0 | 100.0±0.0 | 1.0±0.0 | 23.0±1.5 | 37.1±1.4 | 100.0±0.0 | 0.2±0.0 |
| BUTLER | 1.2B (0.6%) | 100.0±0.0 | 100.0±0.0 | 100.0±0.0 | 1.0±0.0 | 56.5±2.0 | 69.8±2.9 | 100.0±0.0 | 0.5±0.0 |
| LONGMEM | 1.6B (24.6%) | 100.0±0.0 | 100.0±0.0 | 100.0±0.0 | 1.0±0.0 | 58.2±1.9 | 69.9±1.4 | 100.0±0.0 | 0.5±0.0 |
| LM2 | 1.3B (6.3%) | 100.0±0.0 | 100.0±0.0 | 100.0±0.0 | 1.0±0.0 | 57.4±4.0 | 69.6±8.9 | 100.0±0.0 | 0.5±0.1 |
| NESYPR | 1.3B (6.3%) | 100.0±0.0 | 100.0±0.0 | 100.0±0.0 | 1.0±0.0 | **73.5**±3.0 | **80.8**±1.0 | 100.0±0.0 | **0.7**±0.0 |

observations and receives only binary task feedback (success or failure) before proceeding to the next task. NESYPR outperforms the strongest baseline, LongMem, in the Minecraft and Rearrangement domains, achieving improvements of 13.6% in CSR, 11.7% in CGC, and 0.15 in SPL on the test set, thereby demonstrating superior structured and adaptive reasoning capabilities. Similar performance gains are observed in the GlibRearrangement domain as well. More experimental results are provided in Appendix. The single-step planning baselines such as ZSP and RAP, which rely on in-context retrieval-augmented generation [77], show limited reasoning capacity on unseen tasks. The agentic workflow baselines such as CoT, ToT, and GoT, which use reasoning-guidance prompts crafted from the train set [16], perform slightly better, but remain far from achieving reliable task success. The memory-augmented LMs such as LongMem and LM2 surpass the fine-tuning baseline BUTLER, but their average CSR on the test set is still 15.2% lower than that of NESYPR.

**Closed-loop continual embodied tasks.** To further evaluate the generalization performance of NESYPR alongside its adaptability in dynamic settings, we conduct experiments under a closed-loop continual task planning setup in VirtualHome and ALFWorld. The test set is divided into seen and unseen sets, enabling a detailed assessment of the agent's structured and adaptive reasoning capabilities. Unlike open-loop settings, the agent selects actions sequentially in response to intermediate

Table 2: Performance on closed-loop continual embodied tasks in VirtualHome and ALFWorld.

| METHOD | TRAIN | | | SEEN | | | UNSEEN | | |
|---|---|---|---|---|---|---|---|---|---|
| | CSR (↑) | CGC (↑) | SPL (↑) | CSR (↑) | CGC (↑) | SPL (↑) | CSR (↑) | CGC (↑) | SPL (↑) |
| **BENCHMARK: VIRTUALHOME** | | | | | | | | | |
| LLM-PLANNER | 61.0±3.5 | 66.4±2.9 | 0.4±0.0 | 45.5±1.9 | 47.4±2.0 | 0.3±0.0 | 28.8±2.2 | 30.0±2.1 | 0.2±0.0 |
| REACT | 63.6±1.1 | 70.0±0.6 | 0.4±0.0 | 54.7±1.3 | 56.4±0.8 | 0.4±0.0 | 32.7±3.5 | 34.3±3.6 | 0.3±0.0 |
| REFLEXION | 60.8±5.9 | 68.8±5.5 | 0.4±0.0 | 57.2±3.8 | 61.6±5.4 | 0.4±0.0 | 33.7±2.5 | 35.3±2.1 | 0.3±0.0 |
| LONGMEM | 80.5±2.9 | 86.2±2.5 | 0.8±0.0 | 63.3±5.6 | 68.3±6.3 | 0.6±0.1 | 45.7±7.3 | 52.2±8.9 | 0.4±0.1 |
| LM2 | 80.2±4.2 | 85.2±2.9 | 0.8±0.0 | 53.6±5.9 | 57.6±5.2 | 0.5±0.1 | 38.8±4.4 | 43.3±4.4 | 0.3±0.0 |
| DT-MEM | 77.3±3.8 | 80.8±4.8 | 0.7±0.0 | 69.3±5.7 | 71.9±5.9 | 0.7±0.1 | 48.7±5.9 | 52.6±6.0 | 0.5±0.1 |
| OPTIMUS-2 | 79.3±5.4 | 83.9±4.0 | 0.8±0.1 | 70.4±4.4 | 74.0±3.4 | 0.7±0.1 | 44.0±6.1 | 50.9±7.3 | 0.4±0.1 |
| NESYPR | **89.8**±1.9 | **92.3**±1.1 | **0.9**±0.0 | **78.9**±4.5 | **81.7**±2.5 | **0.8**±0.0 | **61.1**±2.2 | **69.3**±2.6 | **0.6**±0.0 |
| **BENCHMARK: ALFWORLD** | | | | | | | | | |
| LLM-PLANNER | 52.4±2.6 | 68.1±2.0 | 0.5±0.0 | 14.0±0.8 | 21.7±0.7 | 0.1±0.0 | 3.3±0.4 | 11.7±0.5 | 0.0±0.0 |
| REACT | 46.3±1.5 | 65.3±1.1 | 0.4±0.0 | 12.8±0.5 | 21.4±0.6 | 0.1±0.0 | 2.9±0.2 | 11.6±0.2 | 0.0±0.0 |
| REFLEXION | 44.3±0.7 | 64.1±0.7 | 0.4±0.0 | 13.0±0.5 | 21.6±0.8 | 0.1±0.0 | 2.9±0.4 | 11.8±0.4 | 0.0±0.0 |
| LONGMEM | 50.1±3.1 | 58.6±2.4 | 0.5±0.0 | 48.6±0.6 | 57.3±0.7 | 0.5±0.0 | 45.2±0.8 | 54.5±0.8 | 0.5±0.0 |
| LM2 | 63.8±1.5 | 71.5±1.4 | 0.6±0.0 | 42.6±2.4 | 46.9±2.0 | 0.4±0.0 | 38.7±2.1 | 45.2±2.5 | 0.4±0.0 |
| DT-MEM | 57.7±2.2 | 61.7±2.7 | 0.6±0.0 | 41.3±2.9 | 48.0±3.3 | 0.4±0.0 | 38.0±3.8 | 44.1±4.2 | 0.4±0.0 |
| OPTIMUS-2 | 59.5±1.5 | 67.4±1.3 | 0.6±0.0 | 52.8±0.9 | 61.6±0.7 | 0.5±0.0 | 49.1±0.7 | 58.7±0.6 | 0.5±0.0 |
| NESYPR | **69.6**±2.7 | **76.2**±2.1 | **0.7**±0.0 | **61.1**±1.1 | **68.6**±1.3 | **0.6**±0.0 | **59.7**±1.4 | **67.9**±1.3 | **0.6**±0.0 |

observations. In Table 2, NESYPR outperforms the strongest baseline in VirtualHome, DT-Mem, with average improvements of 12.5% in CSR and 11.5% in CGC on the train set, 9.6% and 9.8% on the seen set, and 12.4% and 16.7% on the unseen set, respectively. In ALFWorld, NESYPR outperforms the strongest baseline, Optimus-2, achieving average gains of 9.8% in CSR and 8.8% in CGC on the train set, 8.3% and 7.0% on the seen set, and 10.6% and 9.2% on the unseen set, respectively. Notably, the unseen sets show greater performance gains than the seen sets. Combined with an average improvement of 0.12 in SPL, these results indicate that NESYPR performs effective symbolic reasoning. In Appendix, additional results for each incremental configuration are provided, along with complete baseline comparisons. Specifically, both LLM-Planner and agentic workflows such as ReAct and Reflexion exhibit limited capability for symbolic reasoning across benchmarks. While Reflexion leverages past experiences via verbal feedback, it appears to lack the robust reasoning capabilities required in dynamic and complex tasks. Memory-augmented approaches for embodied agents, such as DT-Mem and Optimus-2, outperform other baselines. Yet, NESYPR achieves higher performance, surpassing both DT-Mem and Optimus-2 by an average of 11.2% in CSR and 11.6% in CGC, showing the effectiveness of neurosymbolic proceduralization.

### 5.3 Analysis and Ablation

Table 3: Analysis on proceduralization. LATENCY denotes the agent's planning time in seconds. TOKENS denote the total number of input and output tokens used.

| METHOD | LM | TASK PERFORMANCE | | | REASONING LOAD | | |
|---|---|---|---|---|---|---|---|
| | | CSR (↑) | CGC (↑) | SPL (↑) | LATENCY (↓) | IN TOKENS (↓) | OUT TOKENS (↓) |
| BoT | LLAMA3.1-8B | 53.0±0.5 | 63.5±0.4 | 0.3±0.0 | 59.5±1.9 | 8007.9±103.9 | 1315.4±28.1 |
| | LLAMA3.1-70B | 81.9±0.4 | 85.1±0.3 | 0.6±0.0 | 75.1±3.8 | 7651.0±127.7 | 794.1±33.4 |
| | GPT4.1 | 92.1±0.3 | 93.6±0.2 | 0.7±0.0 | 22.2±2.8 | 7986.1±144.2 | 1202.2±197.2 |
| LRM | DEEPSEEK-R1-8B | 11.5±0.3 | 15.6±0.3 | 0.1±0.0 | 111.0±3.3 | 3198.5±15.8 | 2187.6±69.0 |
| | DEEPSEEK-R1-70B | 26.5±0.4 | 27.5±0.4 | 0.2±0.0 | 209.4±9.2 | 3198.5±15.8 | 1679.3±87.5 |
| | O3-MINI | 78.9±0.4 | 80.8±0.4 | 0.5±0.0 | 18.6±1.7 | 3214.6±15.7 | 2113.9±63.2 |
| PLASMA | LLAMA3.2-1B | 67.4±0.5 | 71.9±0.4 | 0.7±0.0 | 2.7±0.5 | 3221.8±45.3 | 32.7±4.6 |
| | LLAMA3.2-3B | 70.7±0.4 | 75.7±0.3 | 0.7±0.0 | 7.2±0.7 | 3247.7±17.2 | 29.5±5.5 |
| | LLAMA3.1-8B | 80.5±0.5 | 89.2±2.3 | 0.8±0.0 | 18.4±5.8 | 3371.0±13.5 | 122.4±41.6 |
| NESYPR | LLAMA3.2-1B | 73.2±0.4 | 76.0±0.4 | 0.7±0.0 | 1.2±0.3 | 3168.5±0.0 | 30.1±5.3 |
| | LLAMA3.2-3B | 83.6±2.0 | 88.8±2.0 | 0.8±0.0 | 3.5±0.3 | 3169.5±0.0 | 43.6±5.3 |
| | LLAMA3.1-8B | 89.0±2.0 | 93.5±1.8 | 0.9±0.0 | 5.2±0.7 | 3155.5±0.0 | 41.9±6.0 |

**Analysis on proceduralization.** Table 3 presents a comparative analysis of our neurosymbolic proceduralization method to existing proceduralization methods, evaluated in terms of task performance and reasoning efficiency, with a particular focus on enabling timely reasoning through single-step inference. To ensure a fair comparison, we additionally include a unified setting in which all methods are evaluated under identical inference conditions using the same LLaMA 3.1-8B backbone, highlighted in gray background in the table. NESYPR achieves the lowest average plan generation latency, the highest task success rate, and the minimal input and output token usage. BoT and LRM exhibit latencies that are 54.3 and 105.8 seconds longer than NESYPR, respectively, along with 6,125.9 and 2,188.7 more total tokens consumed, due to their reliance on multi-step reasoning. PlaSma, which distills procedural knowledge from larger to smaller LMs, achieves competitive results with efficient inference, reaching 80.5% in CSR using an 8B LM. Yet, NESYPR outperforms it with a higher CSR of 83.6% while operating with only a 3B LM.

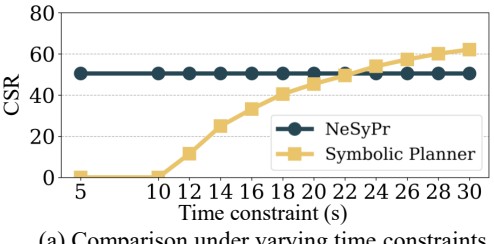

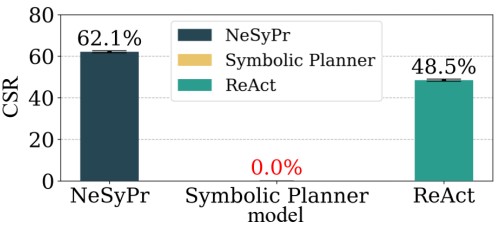

(a) Comparison under varying time constraints

(b) Robustness to dynamic conditions

Figure 3: Comparison with symbolic planner

**Comparison with online symbolic planner.** Figure 3 analyzes the impact of neurosymbolic pro­ceduralization on automated decision-making in agents. Figure 3(a) compares the task success rate of NESYPR and a symbolic planner under a strict time constraint, allowing 1% violation. For tasks where the symbolic planner takes over 10 seconds to find a solution, NESYPR completes planning within a 5-second constraint while achieving 50.6% in CSR. In comparison, the symbolic planner takes up to 22 seconds to reach similar performance. Figure 3(b) evaluates robustness on unseen tasks with dynamic conditions. While the symbolic planner fails when input information is incomplete, NESYPR maintains stable performance and even outperforms ReAct using GPT-4o [78].

Table 4: Analysis on continual embodied task adaptation scenario. During continual task inference, the entire test set is periodically evaluated across 15 intermediate continual evaluation phases.

| METHOD | METRIC | CONTINUAL EVALUATION PHASE | | | | | | | | | | | | | | |
|---|---|---|---|---|---|---|---|---|---|---|---|---|---|---|---|---|
| | | 1 | 2 | 3 | 4 | 5 | 6 | 7 | 8 | 9 | 10 | 11 | 12 | 13 | 14 | 15 |
| LONGMEM | SR (↑) | 57.1 | 56.0 | 58.6 | 54.6 | 47.5 | 51.1 | 52.4 | 51.5 | 52.2 | 52.4 | 53.0 | 53.6 | 55.7 | 54.4 | 53.8 |
| | FWT (↑) | 0.0 | 0.0 | 0.0 | 0.0 | 0.0 | 0.0 | 0.0 | 0.0 | 0.0 | 0.0 | 0.0 | 0.0 | 0.0 | 0.0 | 0.0 |
| | BWT (↑) | 0.0 | 0.0 | 0.0 | 0.0 | 0.0 | 0.0 | 0.0 | 0.0 | 0.0 | 0.0 | 0.0 | 0.0 | 0.0 | 0.0 | 0.0 |
| LM2 | SR (↑) | 64.3 | 52.0 | 55.2 | 57.6 | 52.5 | 53.3 | 50.8 | 50.0 | 49.3 | 48.8 | 49.4 | 50.0 | 52.3 | 52.2 | 51.6 |
| | FWT (↑) | 0.0 | 0.0 | 0.0 | 0.0 | -2.5 | 2.2 | 1.6 | 1.5 | 1.4 | 1.2 | 1.2 | 1.2 | 1.1 | 1.1 | 2.2 |
| | BWT (↑) | 0.0 | 0.0 | 0.0 | 0.0 | 0.0 | -4.4 | -3.2 | -3.0 | -2.9 | -2.4 | -2.4 | -1.2 | -2.3 | -1.1 | -2.2 |
| | FR (↓) | 0.0 | 0.0 | 0.0 | 0.0 | 9.1 | 8.0 | 6.1 | 5.9 | 5.7 | 4.9 | 4.8 | 4.7 | 4.3 | 4.2 | 4.0 |
| | RR (↑) | 0.0 | 0.0 | 0.0 | 0.0 | 0.0 | 0.0 | 0.0 | 0.0 | 0.0 | 0.0 | 0.0 | 2.4 | 0.0 | 2.4 | 0.0 |
| NESYPR | SR (↑) | 64.3 | 64.0 | 62.1 | 63.6 | 65.0 | 66.7 | 61.9 | 60.6 | 60.9 | 61.0 | 61.5 | 61.9 | 63.6 | 62.2 | 61.3 |
| | FWT (↑) | 7.1 | 4.0 | 3.4 | 3.0 | 3.3 | 2.2 | 1.6 | 1.5 | 1.4 | 1.2 | 1.2 | 1.2 | 1.1 | 2.2 | 3.2 |
| | BWT (↑) | 0.0 | 4.0 | 3.4 | 3.0 | 1.7 | 2.2 | 4.8 | 4.5 | 4.3 | 4.9 | 4.8 | 4.8 | 4.5 | 4.4 | 4.3 |
| | FR (↓) | 0.0 | 0.0 | 0.0 | 0.0 | 0.0 | 0.0 | 0.0 | 0.0 | 0.0 | 0.0 | 0.0 | 0.0 | 0.0 | 0.0 | 0.0 |
| | RR (↑) | 0.0 | 12.5 | 10.0 | 9.1 | 7.7 | 7.1 | 13.0 | 12.0 | 11.5 | 12.9 | 12.9 | 12.9 | 12.9 | 12.5 | 12.1 |

**Analysis on continual task adaptation.** Table 4 presents continual adaptation results, highlighting NESYPR's adaptive reasoning using metrics from continual learning [79]. Forward Transfer (FWT) measures how newly acquired procedures improve performance on future tasks by comparing average per-task SR with overall CSR. Backward Transfer (BWT) compares current CSR with that obtained when re-evaluating earlier tasks using retained procedures. Forgetting Rate (FR) is the proportion of previously successful tasks that fail upon re-evaluation, while Recovery Rate (RR) is the proportion of previously failed tasks that later succeed. LongMem, which stores key-value states for retrieval, shows no improvement in FWT and no degradation in BWT. In contrast, LM2, which implicitly maintains a working memory to extend context, shows moderate improvement in FWT but fails to preserve BWT. By leveraging both valid and invalid procedures, NESYPR achieves superior performance in both FWT and BWT. Notably, its FR converges to zero, and it attains about 12.0% RR, demonstrating effective adaptation without symbolic tools.

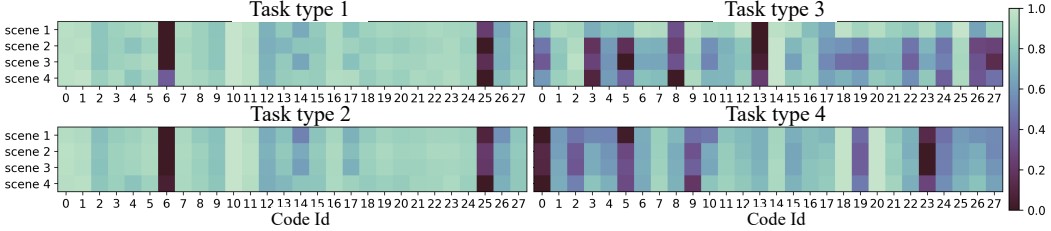

Figure 4: Analysis on procedural memory interpretability

**Analysis on procedural memory interpretation.** Figure 4 shows a heatmap of procedure-unit $c$ (in Eq. (6)) usage across 4 task types and scenes. Task types 1 and 2 share similar solution and exhibit consistent $c$ usage patterns across different scenes. In contrast, task types 3 and 4 involve object-specific actions (e.g., picking up or turning on items), which are more sensitive to scene variations and lead to more divergent patterns.

**Application with different LMs.** Table 5 reports the performance of NESYPR using three LM families across five model sizes [80, 81, 73, 82], evaluated on the Minecraft domain. Across all LMs, NESYPR achieves an average CSR improvement of 10.0% over BUTLER. Performance generally improves with model size, although the degree of improvement varies across different LM families.

Table 5: Application with different LMs

| METHOD | LM | CSR (↑) | CGC (↑) |
|---|---|---|---|
| BUTLER | QWEN2-0.5B | 41.3±1.0 | 54.7±3.3 |
| NESYPR | QWEN2-0.5B | 51.7±1.7 | 60.7±2.9 |
| BUTLER | LLAMA3.2-1B | 51.4±1.9 | 66.7±2.6 |
| NESYPR | LLAMA3.2-1B | 65.2±1.4 | 68.9±2.4 |
| BUTLER | QWEN2.5-1.5B | 50.3±1.5 | 58.6±3.4 |
| NESYPR | QWEN2.5-1.5B | 56.7±1.3 | 59.0±1.5 |
| BUTLER | GEMMA2-2B | 57.5±1.3 | 59.9±0.5 |
| NESYPR | GEMMA2-2B | 66.4±4.1 | 73.8±0.5 |
| BUTLER | LLAMA3.2-3B | 64.2±1.2 | 73.8±2.5 |
| NESYPR | LLAMA3.2-3B | 74.9±2.4 | 77.1±1.8 |

Table 6: Ablation study of NESYPR

| METHOD | CSR (↑) | CGC (↑) |
|---|---|---|
| NESYPR (FULL) | 65.2±1.4 | 68.9±2.4 |
| NESYPR W/O EMA UPDATE OF $\mathcal{C}$ | 63.0±3.1 | 64.8±3.7 |
| NESYPR W/O $\mathcal{C}$ | 47.9±8.4 | 56.5±4.6 |
| NESYPR W/ $\mathcal{M}^+$ ONLY | 63.2±1.3 | 65.1±2.4 |
| NESYPR W/ $\mathcal{M}^-$ ONLY | 63.3±1.4 | 66.4±2.5 |
| NESYPR W/O CP | 59.9±1.4 | 62.9±2.4 |
| NESYPR W/O CP, FOLLOW $\mathcal{M}^+$ | 60.9±1.1 | 64.4±2.2 |
| NESYPR W/O CP, FOLLOW $\mathcal{M}^-$ | 59.7±1.4 | 62.2±2.3 |

**Ablation study.** Table 6 presents an ablation study that evaluates the contribution of each component in NESYPR, using Minecraft. Using the procedure-book $\mathcal{C}$ without EMA updates (i.e., NESYPR W/O EMA UPDATE OF $\mathcal{C}$) results in a performance drop of 2.2% in CSR and 4.1% in CGC. When $\mathcal{C}$ is removed entirely (i.e., NESYPR W/O $\mathcal{C}$), the performance drops by 17.3% in CSR and 12.4% in CGC, demonstrating the importance of a learned $\mathcal{C}$. Models using only successful procedures $\mathcal{M}^+$ (i.e., NESYPR W/ $\mathcal{M}^+$ ONLY) or only failure procedures $\mathcal{M}^-$ (i.e., NESYPR W/ $\mathcal{M}^-$ ONLY)-where the original composed procedure is used as the counterpart in contrastive planning, perform 2.0% and 1.9% worse in CSR, respectively, compared to the full version. Disabling contrastive planning (i.e., NESYPR W/O CP) leads to a further performance drop of 5.3% in CSR. Without CP, simply following either $\mathcal{M}^+$ or $\mathcal{M}^-$ (i.e., NESYPR W/O CP, FOLLOW $\mathcal{M}^+$ and NESYPR W/O CP, FOLLOW $\mathcal{M}^-$) leads to less reliable plan generation, with CSR reductions of 4.3% and 5.5%, respectively.

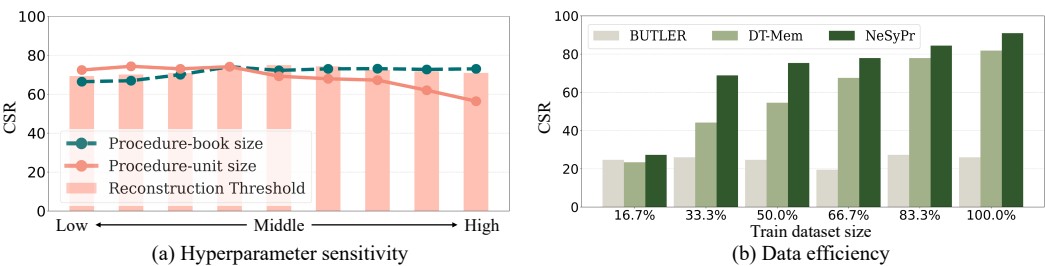

(a) Hyperparameter sensitivity  (b) Data efficiency

Figure 5: Ablation study on procedural memory hyperparameters and learning data efficiency

**Hyperparameter sensitivity and data efficiency.** Figure 5(a) shows that a small size of $\mathcal{C}$ limits task coverage, while a sufficiently large one enables consistently high performance. In contrast, increasing the size of each procedure-unit $c$ reduces diversity in the composed procedures, leading to performance degradation. We also observe that setting a low reconstruction threshold $\upsilon$ causes over-generalization by accepting low-similarity procedures, while a high $\upsilon$ leads to under-generalization by rejecting valid ones. Figure 5(b) shows the data efficiency of each method by reporting task success rates across varying training dataset sizes. NESYPR consistently achieves higher CSR with less data, indicating superior learning efficiency compared to BUTLER and DT-Mem.

## 6 Conclusion

We presented the NESYPR framework that employs neurosymbolic proceduralization inspired by ACT theory. It performs knowledge compilation by abstracting and generalizing multi-step symbolic path-finding and reasoning into single-step inference within an LM. This enables LM-based agents to conduct embodied reasoning efficiently, without relying on large-scale inference engines or online access to symbolic tools. Experimental results on PDDLGym, ALFWorld, and VirtualHome show that NESYPR enables structured, adaptive, and timely reasoning in dynamic embodied environments.

**Limitation and future direction.** As shown in Table 5, NESYPR's performance partially depends on the pretrained knowledge of the LM. To mitigate this, we plan to explore a joint learning strategy that combines knowledge distillation from larger LMs with neurosymbolic proceduralization. This approach has the potential to enhance generalization to more complex, real-world scenarios.

## Acknowledgement

This work was supported by the Institute of Information & communications Technology Planning & Evaluation (IITP) grant funded by the Korea government (MSIT), (RS-2022-II220043 (2022-0-00043), Adaptive Personality for Intelligent Agents, RS-2022-II221045 (2022-0-01045), Self-directed multi-modal Intelligence for solving unknown, open domain problems, RS-2025-02218768, Accelerated Insight Reasoning via Continual Learning, RS-2025-25442569, AI Star Fellowship Support Program (Sungkyunkwan Univ.), and RS-2019-II190421, Artificial Intelligence Graduate School Program (Sungkyunkwan University)), the National Research Foundation of Korea (NRF) grant funded by the Korea government (MSIT) (No. RS-2023-00213118), IITP-ITRC (Information Technology Research Center) grant funded by the Korea government (MIST) (IITP-2025-RS-2024-00437633, 10%), IITP-ICT Creative Consilience Program grant funded by the Korea government (MSIT) (IITP-2025-RS-2020-II201821, 10%), and by Samsung Electronics Co., Ltd.

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
