# OpenReview forum: "NeSyPr: Neurosymbolic Proceduralization For Efficient Embodied Reasoning"
_NeurIPS.cc/2025/Conference — NeurIPS 2025 poster_

### Official Review · Reviewer_fttA · 2025-07-01

**Clarity:** 3
**Significance:** 3
**Originality:** 3
**Rating:** 4
**Confidence:** 2

**Summary:**

This paper introduces NESYPR, a framework that helps language models plan actions in embodied environments without relying on external symbolic planners. It learns from symbolic plans during training and compiles them into reusable procedures stored as vector memories. At test time, the model uses these memories—and feedback from the environment—to adapt and generate plans efficiently. The key idea is to turn multi-step symbolic reasoning into fast, single-step neural inference.

**Questions:**

1. Have you tried training NESYPR on one environment (e.g., PDDLGym) and testing on a structurally different one (e.g., ALFWorld)? This would provide stronger evidence for generalization beyond task-level variation.

2. Given that the unseen tasks come from the same environment as the training tasks, have you considered using few-shot prompting baselines, such as GPT-4 with training examples as prompts?

3. Could you provide more detailed ablations of the contrastive planning component? For instance, what happens when using only successful memory (M+) or changing how memories are retrieved?

4. Do you have any insights into what the procedural memory vectors encode? Are they interpretable or reusable in a symbolic form?

**Ethical Concerns:**

["NO or VERY MINOR ethics concerns only"]

**Limitations:**

While NESYPR shows strong performance within individual environments, its ability to generalize across environments with different symbolic structures remains unclear. The evaluation also lacks comparisons to prompt-based baselines using few-shot examples from training tasks, which could offer competitive alternatives. Additionally, the contrastive planning component, though promising, would benefit from a more thorough ablation to clarify its exact contribution.

**Quality:**

3

**Strengths And Weaknesses:**

Strengths

A major strength of the paper is that it avoids online symbolic planners during inference, yet still achieves strong performance. What's impressive is that NESYPR runs significantly faster than both symbolic planners and large-scale reasoning baselines like BoT or GPT-4, while using much smaller models (e.g., 3B vs. 70B), as demonstrated through latency and token comparisons.

The idea of combining contrastive reconstruction with symbolic plan generation is novel and well-motivated. It offers a promising direction for making planning more adaptive without relying on traditional solvers.

The authors run NESYPR across several popular embodied benchmarks with both seen and unseen tasks. The results are solid and consistent within each environment, suggesting the approach generalizes well to new tasks in familiar domains.

Weaknesses

While the model is evaluated on unseen tasks within the same environment, it is unclear how well it generalizes across environments with different object vocabularies, action schemas, or symbolic representations. A stronger demonstration of generalization would involve training on one environment and testing zero-shot on another.

Since the unseen tasks come from the same environment as the training tasks, it would be natural to construct few-shot prompts using training examples and evaluate models like GPT-4 in an in-context setting. This kind of prompt-based reasoning is a competitive alternative to finetuning or memory-based methods, especially with today's large models. Without such a comparison, it's hard to tell how much of NESYPR's performance comes from its memory mechanism versus simply reusing examples in a smarter prompt.

The contrastive planning mechanism is a core contribution, but the ablation study on it is fairly limited. Table 2 shows results with and without contrastive decoding, but doesn't explore variants (e.g., what happens if only M+ is used, or if memory is retrieved differently). A deeper analysis would help isolate where the performance gains are truly coming from.

---

> ### Author Rebuttal · Authors · 2025-07-27
>
> Dear Reviewer fttA,
>
> Thank you for your thoughtful and constructive review. We appreciate the opportunity to clarify our contributions and address your concerns in detail.
>
> ---
>
> ### **Weakness 1 and Question 1** ###
>
> We appreciate the reviewer's suggestion to evaluate cross-environment generalization. While the unseen tasks originate from the same benchmark, we would like to argue that they already exhibit substantial variation in object configurations, goal predicates, and task structures as described in Appendix C.1.1-C.1.3. These instances introduce entirely novel task types that are not present in the training data, requiring the agent to recombine and reinterpret previously acquired procedural knowledge in a zero-shot manner.
>
> We agree that generalization across environments (e.g., training on a domain in PDDLGym and testing on ALFWorld) is an important direction for future work.
> However, such transfer is non-trivial, as it involves differences in symbolic engine versions and representation formats as shown in Appendix C.1.1-C.1.3.
> Addressing these discrepancies would require additional symbolic alignment mechanisms, which are orthogonal to our primary focus on memory-based proceduralization within a unified neurosymbolic framework.
>
> However, in response to the reviewer's suggestion, we additionally conducted cross-domain evaluation within PDDLGym, where all domains share the same symbolic planner. This setting provides a controlled evaluation for cross-domain generalization without symbolic incompatibility.
>
> |Test Domain|CSR|CGC|
> |-|-|-|
> |Rearrangement (source)|84.9|95.7|
> |Minecraft (target)|20.7|25.4|
> |Glibrearrangement (target)|48.8|82.7|
>
> Interestingly, even within this aligned symbolic setting, transfer across domains (e.g., training on Rearrangement and testing on Minecraft) remains non-trivial.
> This suggests that domain-specific reasoning patterns, such as variations in precondition/effect structures, goal dependencies, and procedural depth, introduce meaningful challenges for cross-domain adaptation.
> Despite the overall drop in performance, we observe that NeSyPr can still generate partially coherent plans in the target domain.
> For instance, in the Minecraft domain, NeSyPr generates domain-specific actions such as `craftplank`, which were never seen during training on Rearrangement, as part of accurate action sequences.
>
> Figure 3 in the main paper illustrates that our evaluation already encompasses scenarios requiring generalization beyond pre-defined domain knowledge. NeSyPr even outperforms symbolic planners under dynamic and partially observable conditions. In settings with incomplete input, symbolic planners often fail due to unmet preconditions, whereas NeSyPr generalizes robustly by retrieving relevant procedures from memory. These results suggest that NeSyPr supports generalization beyond pre-defined domain knowledge by leveraging LM capabilities and grounding knowledge through task experience.
>
> ---
>
> ### **Weakness 2 and Question 2** ###
>
> We fully agree that prompt-based reasoning approaches are strong alternatives to memory-augmented methods, especially when using powerful models such as GPT-4.
>
> As stated in the Introduction, our goal is to perform embodied reasoning efficiently, without online access to large-scale inference or symbolic tools. While prompting large LMs like GPT-4 can yield strong performance in certain settings, these methods typically assume online access, long prompts, and high inference costs. By contrast, our focus is on achieving comparable effectiveness under resource constraints, which is critical for embodied agents operating in latency-sensitive and resource-constrained physical systems.
>
> To evaluate the trade-off between effectiveness and efficiency, as raised by the reviewer, we compared NeSyPr with strong prompt-based methods leveraging large-scale LMs.
> In Table 3 of the main paper, we compare NeSyPr with BoT using GPT-4.1 and LRM using o3-mini. Additionally, we evaluate LLM-Planner, a representative retrieval-augmented in-context learning approach, implemented with GPT-4.1.
>
> |Method|LM|CSR|
> |-|-|-|
> |LLM-Planner (3-shot)|GPT-4.1|81.5|
> |LLM-Planner (8-shot)|GPT-4.1|82.0|
> |BoT|GPT-4.1|92.1|
> |LRM|o3-mini|78.9|
> |NeSyPr|LLaMA3.1–8B|89.0|
>
> NeSyPr achieves performance comparable to or better than much larger prompt-based methods, without requiring extensive prompt engineering, long context windows, or costly inference. In addition, it operates with significantly lower reasoning overhead, including reduced token usage and latency, as detailed in Appendix Table 11.
>
> Additionally, our evaluation includes ten prompt-based approaches:
>
> - Single-step prompting: ZSP, RAP, LLM-Planner
> - Multi-step reasoning: CoT, ToT, GoT
> - Agentic prompting: ReAct, Reflexion
> - Large model prompting: BoT (GPT-4.1), LRM (o3-mini)
>
> These are reported in Tables 1, 2, and 3 of the main paper and described in detail in Appendix C.1.4.
>
> The performance gap between NeSyPr and these baselines, especially under constrained resources, highlights the contribution of procedural memory beyond what can be achieved through prompt reuse alone.
>
> ---
>
> ### **Weakness 3 and Question 3** ###
>
> We agree with the reviewer that a deeper ablation analysis of the contrastive planning mechanism is essential for understanding where performance gains originate. As reported in Table 6 of the main paper, we conducted seven ablation variants that directly address the reviewer's points, including configurations such as using only the successful procedure bank $\mathcal{M}^+$ and modifying the memory retrieval strategy. The table below summarizes key results from Table 6.
>
> |Method Variant|CSR|CGC|
> |-|-|-|
> |NeSyPr (Full)|65.2|68.9|
> |NeSyPr w/ $\mathcal{M}^+$ only|63.2|65.1|
> |NeSyPr w/ $\mathcal{M}^-$ only|63.3|66.4|
> |NeSyPr w/o CP, follow $\mathcal{M}^+$|60.9|64.4|
> |NeSyPr w/o CP, follow $\mathcal{M}^-$|59.7|62.2|
> |NeSyPr w/o CP|59.9|62.9|
>
> In particular:
>
> - **NeSyPr (Full)** uses the complete contrastive planning mechanism with both $\mathcal{M}^+$ and $\mathcal{M}^-$ and the learned procedure-book $\mathcal{C}$.
> - **NeSyPr w/ $\mathcal{M}^+$ only** uses only successful procedures during contrastive planning.
> - **NeSyPr w/ $\mathcal{M}^-$ only** uses only failure procedures during contrastive planning.
> - **NeSyPr w/o CP, follow $\mathcal{M}^+$** disables contrastive planning and directly follows the most similar procedure from $\mathcal{M}^+$.
> - **NeSyPr w/o CP, follow $\mathcal{M}^-$** disables contrastive planning and directly follows the most similar procedure from $\mathcal{M}^-$.
> - **NeSyPr w/o CP** removes contrastive planning entirely.
>
> These configurations, along with other ablations, directly assess the role of contrastive planning as well as the individual contributions of the successful and failure procedure banks.
>
> As summarized in Table 6, contrastive planning yields a 5.3% improvement in CSR, and further removing contrastive planning or relying solely on one type of memory results in additional performance degradation. These findings highlight the importance of contrastive evaluation between positive and negative procedures for robust and adaptive plan generation.
>
> ---
>
> ### **Question 4** ###
>
> Our procedural memory is deliberately designed to encode abstract, reusable task knowledge in a latent vectorized form. Each vector in $\mathcal{C}$ represents procedural abstractions such as precondition–effect dependencies, goal schemas, and operator compositions, in a format that is not directly interpretable but remains functionally composable during inference.
> During inference, the agent retrieves and recombines these vectors based on contextual similarity through contrastive planning, enabling the generation of valid symbolic action sequences without relying on the symbolic planner.
> This design aligns with our ACT-inspired perspective, in which procedural knowledge is retained as implicit production rules that are probabilistically activated depending on the task context.
> As a result, procedural knowledge is represented as quantized latent vectors that implicitly encode condition-action patterns, rather than as explicit symbolic rules. These vectors are compositional, reusable across tasks, and grounded in experience, closely resembling proceduralization in cognitive architectures.
>
> Although individual procedure vectors are not directly interpretable in symbolic form, our behavioral analysis suggests they encode meaningful task structure. As shown in Figure 4 of the main paper, tasks with similar solution strategies consistently activate similar subsets of the procedure-book, even across different environments. In contrast, tasks involving object-specific interactions show more divergent activation patterns, indicating that the procedure-book captures structure relevant to task generalization.
>
> ---
>
> ### **Limitation** ###
>
> We kindly refer the reviewer to our previous responses for clarification on the following points:
>
> - As discussed in our first response, we conducted cross-domain transfer within PDDLGym under a controlled setup. While generalization across different environments is beyond the scope of our current focus, we agree it is a valuable direction for future work.
>
> - As clarified in the second response, we compare NeSyPr with strong prompt-based methods using GPT-4.1 and o3-mini. Despite using significantly smaller models, NeSyPr achieves comparable or better task performance while requiring no prompt engineering, shorter context windows, and significantly lower inference cost. These results highlight that our memory-based proceduralization offers a more efficient and scalable alternative to prompt-based reasoning with large LMs.
>
> - As clarified in the third response, we already provide a detailed ablation study covering seven variants to isolate the contributions of contrastive planning and memory components.
>
> We believe these points have been sufficiently addressed in our responses.

---

> > ### Comment · Area_Chair_4mLt · 2025-08-04
> >
> > Dear Reviewer,
> >
> > Please respond to the rebuttal. Thanks.
> >
> > AC.

---

> > > ### Author Response · Authors · 2025-08-05
> > >
> > > We thank AC for encouraging the discussion. We hope that this response does not violate any rules or expectations for authors.  As the discussion phase is still ongoing, we will make every effort to respond promptly to any reviewer input within a few hours.
> > >
> > > To facilitate a more efficient discussion, we would like to briefly summarize how we addressed the reviewer's feedback:
> > >
> > > - **Cross-environment generalization:**
> > > We acknowledged the importance of evaluating generalization across domains and conducted additional cross-domain experiments within PDDLGym. The results showed that while transfer across domains is non-trivial, NeSyPr can still generate domain-appropriate actions, demonstrating partial generalization. We clarified that differences in symbolic formats across environments (e.g., PDDLGym vs. ALFWorld) require symbolic alignment, which is beyond the current scope but remains a promising direction for future work.
> > >
> > > - **Comparison with prompt-based methods:**
> > > We agreed that prompt-based reasoning is a strong baseline and clarified that NeSyPr achieves comparable or better performance with significantly lower inference cost and no reliance on prompt length or online access, highlighting the benefits of procedural memory. Our evaluation includes GPT-4.1-based methods such as BoT and LLM-Planner.
> > >
> > > - **Ablation study:**
> > > We provided a detailed ablation study (Table 6 in the main paper) covering multiple configurations, including removing contrastive planning, using only positive or negative procedures, and directly following retrieved procedures. The results demonstrate that contrastive planning significantly improves the robustness of plan generation.
> > >
> > > - **Interpretability of procedural memory**:
> > > We clarified that procedural vectors encode implicit condition-action patterns, goal schemas, and operator structures in latent form, aligned with ACT theory. While not directly symbolically interpretable, their functional role is supported by behavioral analysis (Figure 4), where similar tasks consistently activate similar subsets of the procedure-book, indicating meaningful structural encoding.
> > >
> > > We hope these clarifications address the key concerns raised in the review and reflect our commitment to improving the clarity and completeness of the paper.

---

> > > > ### Comment · Area_Chair_4mLt · 2025-08-05
> > > >
> > > > Dear Reviewer,
> > > >
> > > > Please respond to the rebuttal as soon as possible. Thanks.
> > > >
> > > > AC.

---

> > > > ### Comment · Area_Chair_4mLt · 2025-08-07
> > > >
> > > > Dear Reviewer,
> > > >
> > > > Please comment on the rebuttal ASAP. Thanks.
> > > >
> > > > AC.

---

### Official Review · Reviewer_cUia · 2025-07-01

**Clarity:** 2
**Significance:** 3
**Originality:** 3
**Rating:** 5
**Confidence:** 4

**Summary:**

This work introduces a neuro-symbolic framework that distills compositional procedure knowledge from symbolic solvers to a language model. During inference, the system can avoid using symbolic solvers and leverage contrastive planning to generate solutions for unseen tasks. The authors also conduct exhaustive experiments and analysis to demonstrate the effectiveness of their method.

**Questions:**

- I see one potential weakness is the reliance on existing symbolic solvers during training time, which means the system can only solve problems with defined symbols/structure -> It is just making planning more efficient by learning. Have the authors considered distilling LM's knowledge back to the symbolic solver / guide the solver so it could potentially solve more tasks that previously had no existing domain knowledge? Doing this in a cycle to continually evolve the LM seems to be very interesting.
- For PDDL representations, there exist domains where procedural knowledge (efficient imperative rules/feedforward programs if I understand correctly) does not exist, e.g., Sokoban. What is the procedural knowledge learned/computed there?

**Ethical Concerns:**

["NO or VERY MINOR ethics concerns only"]

**Final Justification:**

The direction that this paper is exploring is interesting, and the clarifications have resolved my concerns. I would like to maintain my score.

**Limitations:**

See questions.

**Quality:**

3

**Strengths And Weaknesses:**

**Strength**
- Encoding procedure rules into implicit LM knowledge is an interesting idea, and contrastive planning during inference is also intriguing to me.
- The experimental supports are quite strong, which cover various benchmarks and baselines.
- The problem addressed in this work is important to me, i.e., compositional generalization of neural networks.

**Weaknesses**
The major weakness of this work is the writing issue.
- The experimental section is 4 pages long, making the related works + methodology only 2.5 pages long (with figures). This makes the information in the paper largely biased, and readers can hardly understand the technical details of your algorithm. I'd recommend put some experimental content in the appendix and making more room for the method writing (~3 pages). Here are something I feel confused after trying to carefully read the method:
- What is the input to the LM during training? Is it the plan from the symbolic engine/problem description? What is the specific state space? There should be a running example here.
- How is the procedural knowledge/Rule computed, and what is its format? Is it based on LLM's summarization/ symbolic induction? Please add an example for understanding.
- There are heavy notations across the paper; the authors are expected to provide a notation table.
- Are the unseen tasks from the same domain but with different entity composition? Or are the unseen tasks from different domains but same types?

---

> ### Author Rebuttal · Authors · 2025-07-26
>
> Dear Reviewer cUia,
>
> Thank you for taking the time to review our paper. Your thoughtful and constructive feedback has helped identify areas for improvement. Below, we provide detailed responses to the weaknesses and questions you raised.
>
> ---
>
> ### **Weakness 1** ###
>
> We recognize that the current presentation of the methodology is somewhat compact and that a more detailed ​explanation would improve clarity.
>
> While Sections 2 and 3 in the main paper are relatively compact, we would like to point out that further technical details are included in the Appendix.
>
> - The **related work** is expanded to cover recent advances in agentic frameworks, neurosymbolic methods, and memory-augmented LMs, with comparisons that highlight differences in our approach, as detailed in Appendix A.
>
> - The core **algorithms** are described in detail, including the Compositional NeSy Procedure Learning and Contrastive Planning phases in Appendix B.1.
>
> Since the Appendix was submitted separately from the main paper, we were not able to include explicit cross-references. We will address this in the final version to help readers navigate the supplementary content more easily.
>
> Based on your comment and that of Reviewer sgZ5, we recognize that additional clarification on the ACT theory and its connection to our framework would improve clarity. We will briefly summarize the key components of ACT (e.g., procedural and declarative memory) and clarify how they are implemented in our framework through Compositional NeSy Procedure Learning and NeSy Procedure Contrastive Planning.
>
> While we focused on demonstrating the effectiveness of our framework, we will adjust the allocation to better explain the core components and technical details of our method.
>
> ---
>
> ### **Weakness 2** ###
>
> During the training phase (Compositional NeSy Procedure Learning), as described in Section 3 (lines 110–111), the input to the LM consists of:
>
> - Observation $o$: A symbolic description of the environment state, represented using predicates (e.g., `obj_next_to(plate kitchen_counter)`)
>
> - Goal $g$: A symbolic goal specification (e.g., `(and (open freezer) (obj_inside food_food freezer))`)
>
> - Domain Knowledge DK: Action rules with preconditions and effects that define the action space and dynamics, as shown in Appendix C.1
>
> These inputs are processed by the LM together with working memory, as described in Eq. (3-4) and Appendix Algorithm 1. Representative input examples are shown in Appendix C.1.
> The model is trained to generate action plans that are aligned with the symbolic planner's inputs and outputs.
>
> The state space is defined as the set of all grounded PDDL states. Each state corresponds to a specific truth assignment over the instantiated predicates for the given set of objects in the environment. These symbolic facts represent the current environment configuration in each problem instance.
>
> To illustrate this concretely, Appendix C.1.1-C.1.3 includes some examples for each benchmark:
>
> - PDDLGym: The state includes facts such as `(equipped new-2 agent)` and actions like `equip(log-2 agent)`.
>
> - VirtualHome: The state includes facts like `(holds_lh character tooth_paste)` and actions like `switch_on (character:character, computer:object)`.
>
> - ALFWorld: The state includes relational predicates like `(inreceptacle ?o1 ?r)` or `(iscool ?o)`, and actions such as `pickupobject(agent1:agent, loc_21:location, tomato_1:object, countertop_1:receptacle)`.
>
> All inputs follow PDDL-style representations.
>
> We will expand Appendix C.1 to include additional detailed examples with full symbolic observations, goals, and domain knowledge.
>
> ---
>
> ### **Weakness 3** ###
>
> The procedural knowledge in our framework is not directly summarized by the LM nor handcrafted. Instead, it is learned by encoding the symbolic planner's output (e.g., action plans) into a vector-quantized procedural memory during training.
>
> Specifically, the declarative knowledge used by symbolic tools, such as search algorithms, state transitions, cost estimation, and goal evaluation, is internalized as production rules via a memory-augmented module (Eq. (3-4)) and compressed into discrete procedure-units through vector quantization (Eq. (5-6)). These units form a composite vector, which is stored in a procedure memory (Figure 2).
>
> As a result, procedural knowledge is represented as quantized latent vectors that implicitly encode condition-action patterns, rather than explicit symbolic rules. These vectors are compositional, reusable across tasks, and grounded in experience.
>
> Inspired by ACT theory, these latent procedures can be seen as production rules that are probabilistically activated based on task-specific context. This structure allows our framework to acquire and reuse implicit condition-action mappings through accumulated interaction, closely resembling proceduralization in cognitive architectures.
>
> While these procedures are not symbolically interpretable, we provide an alternative perspective through behavioral analysis. In Figure 4 in the main paper, we visualize the usage frequency of procedure-units across different task types and scenes.
> The results show that tasks with similar solution structure activate consistent subsets of the procedure-book, even across different environments. In contrast, task types involving object-specific interactions lead to more divergent usage patterns.
>
> ---
>
> ### **Weakness 4** ###
>
> We agree that the use of formal symbols throughout the paper could benefit from a centralized summary for easier reference. We will include a notation table summarizing the key symbols and their meanings, as shown below:
>
> |**Symbol**|**Meaning**|**Section**|
> |-|-|-|
> | $\pi_{\text{LM}}$ | LM-based policy for task planning | §2, Eq. (1) |
> | $M$ | Working memory: vector slots for input context | §3.1, Eq. (2) |
> | $\mathcal{C}$ | Procedure-book: codebook of discrete procedure-units | §3.1, Eq (5-6) |
> | $R$ | Runtime procedure: composition of quantized procedure-units | §3.1, Eq. (7) |
> |$\cdots$
>
> ---
>
> ### **Weakness 5** ###
>
> In all benchmarks, unseen tasks belong to the same domain but involve novel goal types or previously unseen combinations of objects and predicates, requiring generalization beyond the training distribution. As detailed in Appendix C.1.2 and C.1.3, in VirtualHome and ALFWorld, unseen tasks introduce entirely new task types, requiring the agent to adapt and generalize based on previously acquired procedural knowledge.
>
> ---
>
> ### **Question 1** ###
>
> Our framework indeed leverages symbolic planners during training to generate optimal plans for proceduralization. As such, it operates in domains where symbolic action rules and typed predicates are defined. However, once training is complete, the model no longer depends on symbolic tools. Instead, it performs adaptive reasoning based on procedural knowledge stored in memory, enabling the agent to reason efficiently without relying on large-scale inference or online access to symbolic tools.
>
> As shown in Figure 3, our model even outperforms symbolic planners under dynamic and partially observable conditions. For example, under tight time constraints, NeSyPr achieves similar or higher success rates while planning significantly faster. In settings with incomplete input, symbolic planners often fail due to unmet preconditions, whereas NeSyPr generalizes effectively by retrieving relevant procedures from memory. These results suggest that proceduralization enables generalization beyond pre-defined domain knowledge by leveraging LM capabilities and grounding knowledge through task experience.
>
> Additionally, we fully agree that enabling the LM to inform or refine symbolic tools in a closed-loop setup is a promising direction. This aligns with recent work, NeSyC [1], which explores continual interaction between symbolic tools and LLMs for constructing generalized symbolic knowledge.
>
> We are planning a follow-up study in this direction. As an initial step, Appendix C.2.3 presents continual learning experiments demonstrating that NeSyPr remains effective across evolving task distributions. Building on this, we aim to extend the framework to continual online environments, where sub-optimal trajectories from the LM agent can be refined by symbolic tools via decomposition or relabeling, and the improved knowledge can be fed back to the agent for further generalization.
>
> [1] Choi, W. et al. "NeSyC: A Neuro-symbolic Continual Learner for Complex Embodied Tasks In Open Domains". ICLR 2025.
>
> ---
>
> ### **Question 2** ###
>
> Sokoban represents one of the most challenging domains for proceduralization, as it requires long-horizon planning, precise spatial reasoning, and backtracking-based solution strategies, with few reusable subgoals or feedforward rules. Small variations in state often lead to completely different solution paths, making it hard to define general procedural abstractions in an interpretable form.
>
> We additionally tested NeSyPr on Sokoban using benchmarks from [2] to assess its performance in procedurally challenging domains. Although the learned procedures are not explicitly interpretable, they capture structural patterns that support generalization across similar Sokoban instances without relying on subgoals or feedforward rules.
>
> |Sokoban|Success Rate|
> |-|-|
> |Train|100%|
> |Test|30.5%|
>
> We note that Sokoban differs from our main focus on embodied domains in that it lacks modularity and procedural structure, making proceduralization particularly difficult. Adapting NeSyPr to such settings may require a distinct approach, and we appreciate the reviewer’s insight in identifying this as a valuable direction for future work.
>
> [2] Su, DiJia, et al. "Dualformer: Controllable fast and slow thinking by learning with randomized reasoning traces". ICLR 2025.
>
> ---
> We hope our responses have adequately addressed your concerns, and we would be happy to provide further clarification if needed. Thank you again for your valuable review.

---

> > ### Comment · Area_Chair_4mLt · 2025-08-04
> >
> > Dear Reviewer,
> >
> > Please respond to the rebuttal. Thanks.
> >
> > AC.

---

> > ### Comment · Reviewer_cUia · 2025-08-06
> > **Response to Authors**
> >
> > Thanks for the detailed explanations and clarifications, and thanks for pointing out the NeSyC paper. I'd like to maintain my score and recommend "accept" of this work. Looking forward to your follow-up work on the continuously evolving version of joint LLM+Symbolic learning.

---

> > > ### Author Response · Authors · 2025-08-06
> > >
> > > Thank you for your thoughtful review and for taking the time to engage in the discussion!
> > >
> > > We truly appreciate your encouraging feedback and your recommendation to accept our work. Your comments have been highly motivating, and we, as authors, see this entire process as an opportunity to grow and pursue even better research in the future. We're looking forward to sharing our follow-up work with you!

---

> ### Author Response · Authors · 2025-08-05
>
> We thank AC for encouraging the discussion. We hope that this response does not violate any rules or expectations for authors.  As the discussion phase is still ongoing, we will make every effort to respond promptly to any reviewer input within a few hours.
>
> To facilitate a more efficient discussion, we would like to briefly summarize how we addressed the reviewer's detailed feedback:
>
> - **Writing clarity and method section length:**
> We recognized the imbalance between the method and experimental sections and will reallocate space in the revised version to better explain key components. We also clarified that detailed algorithms and an extended discussion of related work are provided in the Appendix, and we plan to improve cross-referencing to make this clearer to readers.
>
> - **Input to the LM, state space, and running example:**
> We specified that the LM receives symbolic observations, goals, and domain knowledge in PDDL format. We will further clarify the structure of the state space and provide representative examples across all benchmarks in the Appendix.
>
> - **Procedural knowledge format and learning mechanism:**
> We explained that procedural knowledge is not extracted via summarization but learned through vector quantization of symbolic plans into latent procedure-units. These units implicitly encode condition–action patterns. Their usage across tasks is visualized in Figure 4.
>
> - **Notation overload:**
> We agreed with the suggestion and committed to including a notation table summarizing all key symbols and their definitions.
>
> - **Unseen task definition:**
> We clarified that unseen tasks belong to the same domain but involve novel goals or previously unseen object–predicate combinations, requiring generalization.
>
> - **Reliance on symbolic tools during training:**
> We confirmed that symbolic planners are used only during training to generate optimal plans, while test-time reasoning is fully symbolic-free. We also highlighted that NeSyPr outperforms symbolic planners under dynamic or incomplete conditions and discussed future work on closed-loop LM-to-symbolic feedback.
>
> - **Proceduralization in hard domains like Sokoban:**
> We acknowledged that Sokoban poses a challenge due to its lack of modular structure. We reported results showing that NeSyPr achieves 30.5% generalization on test sets and agreed that adapting proceduralization to such domains is a meaningful direction for future work.
>
> We hope these clarifications demonstrate our efforts to address the reviewer's thoughtful concerns and to improve the clarity and completeness of the paper.

---

### Official Review · Reviewer_La85 · 2025-07-03

**Clarity:** 3
**Significance:** 3
**Originality:** 3
**Rating:** 5
**Confidence:** 3

**Summary:**

The paper presents NESYPR, a neurosymbolic proceduralization based reasoning framework inspired by ACT, which compiles multi-step symbolic reasoning into single-step LM inference. The paper claims this framework eliminates the need for online symbolic planners in embodied tasks. Additionally, the paper develops compositional NeSy procedure learning, which encodes production rules into a vector quantized procedural memory whose vectors can be compositionally combined to generate task specific plans. The paper also implements NeSy procedure contrastive planning, which adaptively generates plans by contrastively reconstructing task-specific procedures from stored procedures labeled with environmental feedback. The paper shows effectiveness and efficiency of NESYPR through extensive
 evaluations on 3 embodied benchmarks and 9 experimental scenarios.

**Questions:**

None

**Ethical Concerns:**

["NO or VERY MINOR ethics concerns only"]

**Final Justification:**

Thanks to the authors for addressing my concerns. The new experiment results presented in rebuttal address my concerns. I'd recommend authors to add this to the main paper. I have updated my rating to reflect the same.

**Limitations:**

The proposed method is making a non-standard change to the LLM architecture which likely leads to catastrophic forgetting of pretrained knowledge of the LLM which limits applicability of this method to more general problems and domains.

I think the paper itself is interesting and has a bunch of interesting experiments. I am willing to increase my rating if authors can clarify the questions I asked in weaknesses section and add additional ablations.

**Quality:**

3

**Strengths And Weaknesses:**

Strengths:

1. The paper is well written and easy to follow
2. The proposed method outperforms strong baselines on all the benchmarks and tasks that authors consider for task planning in the paper which shows effectiveness of the method.
3. Additionally, the method is also efficient at inference time and performs effectively under a time constraint compared to symbolic planners.
4. The analysis in figure 4 also shows neat results where procedure codebooks across tasks show shared features
5. The ablations provided in the paper are also thorough and valuable to show effectiveness of the method.
6. Additionally, the analysis in figure 5 is also insightful. The finding that increasing codebook size leads to drop in composition performance is a interesting finding.

Weaknesses:

1. In Table 3 why does Deepseek 8B model (row 4) have a higher latency at almost 1/2 number of tokens compared to llama 3.1 8b (row 1)? This seems counterintuitive. Can authors explain why this is the case? Instead of latency isn’t it better to compare efficiency by FLOPs? Can author show similar analysis using FLOPs?
2. One thing that is a bit unclear is the effectiveness of pretrained representations which come from using a pretrained language model when learning the procedural memory for neurosymbolic planning. I understand the motivation to equip LLMs with neurosymbolic planning ability with procedural memory but the current experiments do not ablate the effectiveness of using a pretrained LLM as initial starting weights vs using a LLM from scratch to learn such neurosymbolic planning. Currently, it is unclear whether the gains in performance are coming from pretrained representation + architecture change vs just the changes in architecture and training recipe. I would appreciate it if authors can run this experiment as I believe it will make the paper stronger.

---

> ### Author Rebuttal · Authors · 2025-07-27
>
> Dear Reviewer La85,
>
> Thank you for your thoughtful and constructive review. We appreciate the opportunity to address your questions in detail and provide additional analysis to clarify our contributions.
>
> ---
>
> ### **Weakness 1** ###
>
> We appreciate the reviewer's observation.
> The latency difference between BoT (LLaMA3.1-8B) and LRM (DeepSeek-R1-8B) in Table 3 of the main paper may seem counterintuitive, but it becomes clear when comparing the input and output token lengths, as shown in Appendix Table 11. For reference, the evaluation results from Appendix Table 11 are reproduced in the table below.
> BoT uses long prompts as input, leading to many input tokens but relatively short outputs.
> In contrast, LRM starts with short prompts but generates a much larger number of output tokens due to its multi-step reasoning.
>
> In decoder-only models, input tokens are processed in a single forward pass, while output tokens are generated sequentially via autoregressive decoding. As a result, output length becomes the dominant factor in inference latency.
> In the case of LRM, although it uses shorter input prompts than BoT, it generates significantly more output tokens due to its multi-step reasoning strategy.
> As a result, the higher number of generated tokens in LRM leads to substantially higher plan generation latency.
>
> To provide a hardware-independent measure of efficiency, we agree that reporting the computational cost in terms of floating point operations (FLOPs) is more appropriate.
> Accordingly, we report the FLOPs for BoT, LRM, PlaSma, and NeSyPr using the LLaMA3.1-8B backbone, measured with the official PyTorch profiler on an NVIDIA A100 server.
> The profiler was applied to plan generation runs using task inputs from the same benchmark as in Table 3 of the main paper. The results are summarized in the table below.
>
> |Method|CSR (%)|Latency (s)|In Tokens|Out Tokens|FLOPs|
> |-|-|-|-|-|-|
> |BoT|53.0|59.5|8007.9|1315.4|86.8+$\alpha$ TFLOPs|
> |LRM|11.5|111.0|3198.5|2187.6|55.4 TFLOPs|
> |PlaSma|80.5|18.4|3371.0|122.4|236.9 TFLOPs|
> |NeSyPr|89.0|5.2|3155.5|41.9|98.3 TFLOPs|
>
> While NeSyPr incurs slightly higher FLOPs than simpler baselines such as LRM, this overhead stems from its memory-augmented module and contrastive planning mechanisms, which are designed to enhance robustness and correctness in reasoning.
> It is important to note that in our implementation, the problem distillation step of BoT is performed by GPT-4.1 regardless of the base LM, as described in Appendix C.1.4. Consequently, this additional step is excluded from FLOPs measurement, and we denote it as "+$\alpha$" to reflect the unaccounted cost—potentially underestimating the actual computational burden of BoT.
> In contrast to prompt-based or distillation-based approaches, NeSyPr actively composes and validates procedures during inference, trading off minimal compute overhead for significantly higher planning accuracy and consistency.
>
> Despite this moderate FLOPs usage, NeSyPr achieves the best trade-off across success rate, latency, and token efficiency. This demonstrates that structured reasoning can remain performant without relying on excessively large models or token-intensive decoding.
>
> Furthermore, as reported in Appendix Table 12, NeSyPr maintains fast inference times (e.g., 5.0s on Jetson Orin) and stable performance even on embedded devices, confirming that the compute overhead remains practical for real-world deployment. The table below references the same experimental results from Appendix Table 12.
>
> |Device|Latency Min (s)|Latency Max (s)|Latency Avg (s)|Mem Min (GB)|Mem Max (GB)|Train SR|Train GC|Seen SR|Seen GC|Unseen SR|Unseen GC|
> |-|-|-|-|-|-|-|-|-|-|-|-|
> |RTX A6000|0.5|5.1|1.5|0.0|1.9|97.4|97.7|88.4|89.6|61.5|67.9|
> |RTX 4090|0.5|4.5|1.0| 0.0|1.9|96.1|98.7|88.4|89.7|61.5|68.5|
> |RTX 3090|0.5|4.8|1.2|0.0|1.9|92.2|94.6|86.6|87.8|61.5|68.5|
> |RTX 3050|0.6|7.7|1.6|0.0|1.9|94.8|96.9|84.8|86.5|61.5|67.9|
> |Jetson Orin|2.0|11.4|5.0|4.3|8.6|93.5|96.1|87.5|88.7|59.6|66.6|
>
> Due to the inherent overhead of profiling, we were unable to replicate the full-scale evaluation for all settings in Table 3 of the main paper, but we plan to extend this analysis with more comprehensive measurements in the final version.
>
> ---
>
> ### **Weakness 2** ###
>
> We fully agree that disentangling the effects of pretrained representations and architectural design is essential to understanding the source of performance gains in NeSyPr.
>
> Our implementation adopts a parameter-efficient LoRA-based finetuning [1] setup, which preserves the pretrained weights and updates only lightweight adapter layers, as described in Appendix C.1.4. This design choice was made to address the low-data regime, where full finetuning or training from scratch may lead to instability or inefficiency. We clarify this rationale more explicitly in the revised version.
>
> Additionally, to directly address the reviewer's concern, we conducted an ablation study comparing three LM training configurations: (1) training from scratch, (2) full finetuning, and (3) LoRA-based finetuning.
> Each configuration was evaluated both with and without the NeSyPr architecture, using the LLaMA‑3.2‑1B model on the PDDLGym benchmark. The results are presented below:
>
> |Domain|\||LM-Scratch||\|||LM-FullFinetune||\|||LM-LoRA||\|||NeSyPr-Scratch||\|||NeSyPr-FullFinetune||\|||NeSyPr-LoRA||\|||
> |-|-|-|-|-|-|-|-|-|-|-|-|-|-|-|-|-|-|-|-|-|-|-|-|-|-|
> ||\||SR|GC|\|||SR|GC|\|||SR|GC|\|||SR|GC|\|||SR|GC|\|||SR|GC|\|||
> |Minecraft|\||0.3|39.4|\|||2.0|31.2|\|||75.7|78.4|\|||0.4|57.5|\|||6.1|39.6|\|||82.6|84.5|\|||
> |Rearrangement|\||2.6|29.3|\|||5.0|44.3|\|||78.3|84.9|\|||3.0|45.6|\|||20.0|49.8|\|||86.8|90.4|\|||
> |Glibrearrange|\||0.0|39.5|\|||1.2|49.0|\|||73.3|78.6|\|||2.4|40.4|\|||10.3|55.4|\|||79.4|86.2|\|||
>
> Across all training configurations, including the from-scratch setting, the addition of procedural memory consistently improves performance and highlights the effectiveness of structured memory-based reasoning.
> With the architecture fixed, using pretrained weights improves NeSyPr's performance over training from scratch, highlighting the value of pretrained representations in proceduralization.
> Given the same pretrained initialization, augmenting the model with procedural memory (i.e., transitioning from LM-LoRA to NeSyPr-LoRA), as well as preserving pretrained weights while adding procedural memory (i.e., transitioning from NeSyPr-FullFinetuning to NeSyPr-LoRA), leads to further performance gains. This suggests a synergistic effect between pretrained representations and procedural memory.
>
> We believe this ablation further supports the complementary roles of pretrained representations and procedural memory in NeSyPr.
>
> [1] Hu, Edward J., et al. "Lora: Low-rank adaptation of large language models." ICLR 2022.
>
> ---
>
> ### **Limitation** ###
>
> While we acknowledge that catastrophic forgetting is a well-known challenge in continual learning and LM finetuning, we clarify that our method introduces no parameter modification to the backbone LM. NeSyPr employs a parameter-efficient LoRA-based finetuning strategy that keeps the pretrained weights frozen, while augmenting the model with an external procedural memory module. This design ensures that the pretrained knowledge within the LM remains intact throughout both training and inference.
>
> Empirically, we observe that the LLaMA-3.2‑1B backbone retains its pretrained capabilities, with no change in perplexity on WikiText-103 before and after training (16.4 ppl). Since the procedural memory operates as a complementary reasoning module that encodes symbolic abstractions of previously successful solutions, it guides planning without modifying the LM's internal parameters.
>
> Additionally, our framework demonstrates robustness in both continual adaptation and continual learning scenarios.
> As shown in Table 4 of the main paper, NeSyPr achieves strong performance in the continual adaptation scenario, maintaining stable success rates over 15 evaluation phases, with positive forward and backward transfer, zero forgetting, and consistent recovery.
> In a separate continual learning scenario, Appendix Figure 5 shows that NeSyPr-Incremental, which gradually expands procedural memory over time, achieves steady improvement across 10 learning phases and eventually approaches the performance of an oracle trained on all tasks at once.
> Together, these results confirm that our method not only avoids catastrophic forgetting of pretrained knowledge but also enables continual retention and reuse of experience in dynamic embodied environments.
>
> ---
>
> We have carefully addressed the concerns raised in the weaknesses section and included additional ablation studies as recommended. We greatly appreciate your thoughtful review and hope these updates provide a clearer understanding of our contributions.

---

> > ### Comment · Area_Chair_4mLt · 2025-08-04
> >
> > Dear Reviewer,
> >
> > Please respond to the rebuttal. Thanks.
> >
> > AC.

---

> > > ### Author Response · Authors · 2025-08-05
> > >
> > > We thank AC for encouraging the discussion. We hope that this response does not violate any rules or expectations for authors.  As the discussion phase is still ongoing, we will make every effort to respond promptly to any reviewer input within a few hours.
> > >
> > > To facilitate a more efficient discussion, we briefly summarize how we have addressed the reviewer's feedback:
> > >
> > > - **On latency vs. tokens (Table 3):** We clarified that the latency difference between BoT and LRM arises from their output token lengths, as autoregressive decoding makes output length the dominant factor in inference time. We also conducted a FLOPs-based analysis using PyTorch profiler, showing that NeSyPr achieves the best trade-off across success rate, latency, and compute cost. The results were added alongside device-level profiling, including an embedded device.
> > >
> > > - **On the role of pretrained representations:** We conducted an ablation study comparing models trained from scratch, with full finetuning, and with LoRA-based finetuning, both with and without NeSyPr. The results demonstrate that NeSyPr consistently improves performance, and that pretrained weights further enhance proceduralization, confirming the complementary role of architecture and initialization.
> > >
> > > - **On catastrophic forgetting:** We clarified that NeSyPr uses LoRA-based finetuning and keeps the backbone LM weights frozen. We empirically verified that the model retains its pretrained capabilities (e.g., unchanged perplexity on WikiText-103) and demonstrated continual adaptation and continual learning performance, showing no forgetting and strong knowledge retention.
> > >
> > > We hope these additions clarify the key points raised in the review and show our commitment to improving the clarity and completeness of the paper.

---

> > > > ### Comment · Area_Chair_4mLt · 2025-08-07
> > > >
> > > > Dear Reviewer,
> > > >
> > > > Please comment on the rebuttal ASAP. Thanks.
> > > >
> > > > AC.

---

### Official Review · Reviewer_sgZ5 · 2025-07-03

**Clarity:** 1
**Significance:** 2
**Originality:** 3
**Rating:** 4
**Confidence:** 3

**Summary:**

In my understanding, the paper proposes an approach for task planning by combining transformer-based models and symbolic solvers. First a transformer model is trained to generate actions that solve a given problem definition, supervised by the solution of the PDDL solver. The transformer is augmented by a procedure codebook, an explicit memory with latent codes that are used to generate the actions. Then, the pretrained LM samples a set of latent procedures and select the procedures that maximize the likelihood of the transformer model. The approach is tested in 3 different environments, modifying existing benchmarks to encode sequences of tasks. The proposed method improves over several LLM-based and symbolic planning based approaches, obtaining higher success rate and lower computation time.

**Questions:**

- Provide one example to ground the method. State what are the inputs of the LM, and how is M formed to generate the procedural memory R.
- Better interpretability of what the codebook R is doing. Would it be possible to decode the R codes back into task predicates?
- How does Equation 1 and 9 relate to each other?
- How are the positive and negative procedures obtained in the contrastive planning stage?
- How does the approach compare to works like DualFormer? (https://arxiv.org/abs/2410.09918)

**Ethical Concerns:**

["NO or VERY MINOR ethics concerns only"]

**Final Justification:**

Based on the author's rebuttal as well as other reviewer's points, the contributions of the proposed method, and the justification of their approach are much clearer to me. I still think that clarity could be greatly improved, but I think that the paper merits acceptance, hence I change my score to borderline accept.

**Limitations:**

Yes

**Quality:**

2

**Strengths And Weaknesses:**

Strengths:
- Very thorough experiments and strong results. The method is compared in 3 different benchmarks in with a large number of baselines. The method obtains significant improvements across the 3 benchmarks compared to both LM-based baselines as well as baselines using symbolic planners. Furthermore, the latent procedures can be reused across tasks, leading to the agent being able to adapt as goals change in the environment, compare to other baselines.
- Ablations of the effect of different LLMS to generate the symbolic predicates, as well as the effect of the different components of the NeSYPR approach.

Weaknesses:

Clarity: In the current state the paper is too difficult to follow to be a valuable contribution for a conference. I would strongly advice to remove abstract statements that do not provide information about the method (e.g. grounding parts of the method on human knowledge) and try to ground the framework in a concrete problem earlier on. The method is stated too abstractly and it is really difficult to follow. Below more details.

- The paper uses a lot of concepts without clear meaning. Why is "multi-step symbolic structured path-finding and reasoning into single-step LM inference, akin to human knowledge compilation"? What does "triggered automatically as cognitive procedures" (l42) mean? The second paragraph in the introduction doesn't provide any clarity on how declarative vs procedural memory are used in this framework, a concrete example to ground the framework would help improve clarity. The introduction is too abstract and it is really hard to understand what the method is solving or why it enables faster LM inference.
- The methods in line 64 to 72 are referenced without further context which makes it hard to understand why the stated results are relevant.
- Equation 1 is introduced without any explanation or reasoning. Why do we want to minimize the KL divergence between the LM policy and the tool policy? Are the outputs of those policies in the same domain?
- Figure 1 includes a lot of details that are barely described in the main text.
- Overuse of notation. In equation 2, is the M in the input the same as the M at the output of the decoder block? Is E_self the same as H_{l-1} ? What is M for the first layer?
- What makes a procedure positive or negative in Equation 10? How are the procedures R+ and R- constructed?

Method:
- The method seems overly complex, and doesnt justify some of the decisions. Why learning a codebook end to end with the planning? The general approach is to learn these in two stages. Does this lead to codebook collapse? Did authors follow the more common approach? Why isn't there a reconstruction loss in the VQ loss? The loss in equation 9 does not match the equation 1, what is the relationship between both equations?
- Is the LM based agent represented in Figure 2? What makes that model a language model? Is it initialized from a pretrained LM? From the text and figure it looks like a transformer.
- How does the proposed method enable planning under dynamic environments, particularly in changing dynanmics and goals over time? What part of the model is enabling that?
- If I understand correctly, the input to the NeSyPR model is a PDDL like definition of the current problem and the domain. Where does that come from? How is it extracted from an environment? Is an LLM used to generate that?
- Why does the process in section 3.2  supress failure patterns? (L169).

Results:
- Beyond the quantaitive results, it is very hard to evaluate why the propsoed method is doing better. Showing some qualitative examples or understanding how the propsoed approach is performing neurosymbolic reasoning would be helpful.
- What do the evaluation phases represent in Table 4?


- Related work: Some form of related work should be included in the main paper, it is really hard to understand where the presented work fits in the literature. It seems like the paper fits into the second family of approaches (neurosymbolic methods that integrate LMs) but it is unclear what the main differences are.

---

> ### Author Rebuttal · Authors · 2025-07-28
>
> Dear Reviewer sgZ5,
>
> We appreciate the reviewer's feedback. Below, we provide clarifications and responses to the suggestions.
>
> ---
> ### Clarity weakness ###
> **w1.** NeSyPr transforms multi-step symbolic reasoning into single-step neural inference, substantially reducing inference latency. This process mirrors human proceduralization as described in ACT [1], where repeated experience converts declarative knowledge into fast-access production rules.
>
> In ACT, production rules (condition–action mappings) are automatically triggered when their conditions match the contents of working memory. Similarly, in NeSyPr, once the agent's working memory encodes the current observation and goal, relevant procedures are retrieved from memory based on similarity (e.g., cosine $\geq$ 0.95), without requiring external prompting.
>
> NeSyPr separates declarative and procedural knowledge in both representation and execution. During training, declarative knowledge such as search algorithms, transitions, and goal evaluation is used to generate symbolic plans, which are then encoded into vector-quantized procedural memory. At test time, the LM bypasses symbolic reasoning and generates plans by attending to the procedural memory, enabling single-pass inference.
>
> [1] John R Anderson. Cognitive science series. The architecture of cognition. 1983.
>
> **w2.** Lines 64-72 of the Introduction summarize NeSyPr's empirical contributions, organized around the core challenges of structured, adaptive, and timely reasoning. Each cited baseline highlights a specific comparison point: DeepSeek-R1-Distill for structured reasoning, the symbolic planner for adaptive reasoning, and BoT for timely reasoning. In the final version, we plan to enrich this overview by incorporating the detailed baseline descriptions provided in Appendix C.1.4.
>
> **w3.** Eq. (1) includes an imitation learning objective [2], minimizing the KL divergence to align the LM policy with the symbolic planner. As both operate over the same symbolic action space, their outputs are directly comparable.
>
> [2] Wonje Choi et al. Embodied CoT Distillation From LLM To Off-the-shelf Agents. ICML 2024.
>
> **w4.** Figure 1 illustrates the two main phases of NeSyPr: i) Compositional NeSy Procedure Learning and ii) NeSy Procedure Contrastive Planning. The roles and workflows of each phase are outlined in Section 3 (Lines 105-120), and we plan to enhance this explanation by incorporating additional implementation details and algorithmic steps from Appendix B.
>
> **w5.** While we recognize that the notation is dense, we do not believe it constitutes overuse. As addressed in our response to Reviewer cUia's Weakness 4, we will include a notation table in the Appendix to enhance clarity and readability.
>
> In Eq. (2), working memory $M$ is updated at each layer and passed forward, despite sharing the same symbol. Self-attention output $E_\mathrm{self}$ is computed by applying Self-attention to the previous layer's hidden state $H_{l-1}$ and is not identical to $E_\mathrm{self}$. For the first layer, $M$ is initialized using the output from the final layer of the previous inference step.
>
> **w6.** As described in Section 3.2 of the main paper and Algorithm 2 in Appendix B.1, the agent maintains two procedure banks: $\mathcal{M}^+$ for successes and $\mathcal{M}^-$ for failures. During inference, each $c_i \in R$ is compared against both banks using a similarity threshold (e.g., $\upsilon = 0.95$). If a match is found, $c_i$ is replaced; otherwise, it remains unchanged. This runtime produces $R^+$ and $R^-$ for contrastive planning. After each task, the current procedure is added to $\mathcal{M}^+$ or $\mathcal{M}^-$ based on binary feedback.
>
> ---
> ### Method weakness ###
> **w1.** The goal of our study is to enable the LM to generate action sequences guided by a symbolic planner via single-step inference. This requires end-to-end training of the inputs, memory module, LM, and output sequence [3,4]. In continual learning settings, as in Appendix C.2.4, the learning of the memory module and LM can be decoupled, allowing each to be trained incrementally across stages [5].
>
> To prevent collapse of the procedure-book $\mathcal{C}$, we apply stabilization techniques beyond the commitment loss in Eq. (8), including periodic reinitialization of the $\mathcal{C}$ and careful tuning of EMA hyperparameters. As shown in Table 6, the 'w/o EMA update of $\mathcal{C}$' variant reduces CSR and CGC by 2-4%, but avoids the collapse.
>
> Unlike conventional VQ-based methods such as VQ-VAE that rely on pixel-level reconstruction losses, our approach optimizes task performance via the behavior log-likelihood term in Eq. (9).
>
> Eq. (1) defines the ideal policy objective: maximizing task success while minimizing KL divergence from the symbolic planner. In practice, we approximate this by supervising the LM using planner-computed action sequences and symbolic inputs.
>
> [3] Weizhi Wang et al. Augmenting language models with long-term memory. NeurIPS 2023.
>
> [4] Jikun Kang et al. LM2: Large Memory Models. arXiv 2025.
>
> [5] Jikun Kang et al. Think before you act: Decision transformers with working memory. ICML 2024.
>
> **w2.** Figure 2 shows the architecture of NeSyPr's LM-based agent, with components laid out from left to right. The 'Language Model' in Figure 2 includes a decoder-only Transformer with Decoder Blocks, Self-Attention, and Feed-Forward layers, generating a token sequence as the action plan. As described in Appendix B.2, NeSyPr builds on a pretrained LM and incorporates a memory-augmented module. Table 5 of the main paper demonstrates its compatibility across various pretrained LM families and model scales.
>
> **w3.** In a continual task stream, NeSyPr enables planning in dynamic environments by leveraging contrastive planning, which allows the agent to adaptively reconstruct procedures based on environmental feedback. Using stored experiences of prior successes and failures, the agent applies contrastive planning to guide generation toward reliable behaviors while suppressing failure patterns. As this process repeats, the procedure banks are continually updated, enabling retrieval and refinement of appropriate procedures within a single forward pass, even under changing conditions. This adaptive reasoning capability is demonstrated in Table 4 and Figure 3(b) of the main paper.
>
> **w4.** We directly use benchmarks that provide symbolic PDDL-style definitions, such as PDDLGym, VirtualHome, and ALFWorld, where the domain and problem descriptions are already structured for symbolic planners. No LLM is used to extract or generate these definitions. This design is noted in Section 4.1 of the main paper and detailed in Appendix C.1.1-C.1.3.
>
> **w5.** At each decoding step, NeSyPr computes token distributions conditioned on successful and failed procedures ($p^+$ and $p^-$) and derives a contrastive score by subtracting $p^-$. This suppresses failure-prone tokens and lowers the chance of repeating invalid plans.
>
> ---
> ### Results weakness ###
> **w1.** In Figure 4 of the main paper, we visualize procedure-book usage patterns as a heatmap to illustrate how different procedure-units are composed and reused across four task types. Tasks with similar logics activate overlapping units, while object-specific tasks use more diverse ones. We also plan to include side-by-side examples of successful and failed plans to help clarify the sources of performance gains.
>
> **w2.** Table 4 evaluates how the agent adapts over time in a continual task setting. We measure forward transfer (benefits to future tasks) and backward transfer (impact on past tasks) as indicators of adaptation. The agent solves tasks sequentially, and the full test set is periodically re-evaluated across 15 checkpoints to reflect accumulated procedural knowledge.
>
> **w3.** We acknowledge that the Related Work section in the main paper is a condensed summary of the full discussion in Appendix A, where prior work is grouped into three areas: (i) agentic frameworks, (ii) neurosymbolic methods, and (iii) memory-augmented LMs, and NeSyPr is positioned in contrast to each. In the final version, we plan to expand this section by incorporating key theoretical background, including ACT theory, into the main paper.
>
> ---
> ### Questions ###
> **q1.** The input includes observation, goal, and domain knowledge, all represented as PDDL strings (examples are in Appendix C.1). The working memory $M$ encodes the current state via memory-augmented cross-attention, and the runtime procedure $R$ is obtained by vector-quantizing $M$ using the procedure-book $\mathcal{C}$.
>
> **q2.** $R$ is derived from the procedure-book $\mathcal{C}$, which encodes implicit production rules as composable latent representations. While not symbolically interpretable, $R$ integrates seamlessly into LM inference. We provided behavioral analysis in Figure 4 of the main paper to offer insight into its role.
>
> **q3.** Please refer to our response to Method weakness w1.
>
> **q4.** Please refer to our response to Clarity weakness w6.
>
> **q5.** We thank the reviewer for suggesting this relevant baseline. Although DualFormer was published recently (ICLR 2025), it is a meaningful baseline in the context of proceduralization.
> NeSyPr is inspired by ACT, compiling declarative knowledge into procedural memory and adaptively reconstructing it for reasoning in embodied environments. In contrast, DualFormer follows a System-1/2 framework, learning from randomly sampled reasoning traces to train a single Transformer that dynamically adjusts its reasoning depth based on task difficulty.
> We provide a comparison between DualFormer and NeSyPr in the Minecraft environment.
>
> |Method|Mode|Train CSR|Train CGC|Test CSR|Test CGC|Latency (s)|
> |-|-|-|-|-|-|-|
> |NeSyPr||100.0|100.0|65.2|68.9|4.7|
> |DualFormer|Fast|100.0|100.0|52.2|63.2|5.2|
> ||Slow|100.0|100.0|55.0|65.8|6.2|
> ||Auto|100.0|100.0|55.3|64.2|5.8|
> ---
> We hope our responses clarify the contributions of our work.

---

> > ### Comment · Reviewer_sgZ5 · 2025-08-04
> > **Response to authors**
> >
> > Thank you for the detailed response. The rebuttal has clarified most of my questions. I appreciate the extra baseline despite how recent DualFormer is. While I think that the clarity of the paper could still significantly be improved, based on the rebuttal and other reviewers comments I am changing my score to borderline accept. I would strongly encourage making the updates referred in the rebuttal in the final version. A few comments and suggestions:
> >
> > Clarity:
> > - *W2*: The motivation for the different baselines was clearer in the response. I would suggest, for the final version, including a short explanation on what properties each baseline is comparing, to justify the choice of the baselines.
> > - *W3*: How is the KL divergence computed here? My understanding is that the policy \PI_{tool} is deterministic. Is this policy only 1 for the right tool, given o_t, g?
> > - *W5*: I would recommend using two different symbols or indices for the M at the input and at the output.
> >
> >
> > Method:
> > - *W1*: Thanks for the clarifications, I do understand better the design choices. I don't think the log-likelihood term is different from conventional VQ-VAE methods (pixel loss is an instance of log-likelihood). I would appreciate if it was clearer in the main text that Eq. 1 is an ideal policy, and why Equation 9 is a good approximation (e.g. why we can remove the KL divergence term).
> >
> >
> > Results:
> > - *W1*: Thank you for the comment. I think that a side by side example of successful and failed plans will help clarify the performance gains.

---

> ### Author Response · Authors · 2025-08-05
>
> We sincerely thank the reviewer for initiating the discussion on our submission, for the thoughtful follow-up, and for reconsidering the evaluation based on our rebuttal and clarifications!
>
> While we have done our best to provide additional clarifications in the rebuttal, we fully acknowledge that the reviewer's feedback highlights points that should be explicitly addressed in the revised version. Depending on the final decision and guidelines for final version, we plan to revise the main paper to incorporate not only the common concerns raised across reviews but also the issues we ourselves identified as unclear during the rebuttal process. These revisions will be prioritized within the allowed space in the main paper. We also commit to including all remaining clarifications, along with relevant content from the review, rebuttal, and discussion phases, in the Appendix.
>
> Our response to the reviewer's follow-up comment in the discussion is as follows:
>
> ### Clarity: ###
>
> - *W2*: Thank you for the suggestion. In the revised version, we will explicitly clarify what properties each baseline is selected to highlight, thereby strengthening the justification for our empirical comparisons.
>
>     - **DeepSeek-R1-Distill** [1] is included to highlight NeSyPr's structured reasoning performance. Although DeepSeek-R1-Distill (70B) leverages internalized reasoning chains and benefits from large-scale model capacity, it still underperforms NeSyPr in task success rate, despite having nearly 70 times more parameters.
>     - **Symbolic planner** [2] is included to highlight NeSyPr's adaptive reasoning performance. While the symbolic planner enables precise inference through domain knowledge, it lacks the flexibility to adapt in dynamic or partially observable environments.
>     - **Buffer of Thoughts (BoT)** [3] is included to highlight  NeSyPr's timely reasoning performance. BoT achieves high task success through iterative insertion of CoT templates, but it suffers from significant token and latency overhead due to multi-step generation.
>
> - *W3*: We thank the reviewer for the thoughtful question. In Eq. (1), we adopt a standard imitation learning objective by minimizing the KL divergence between the LM policy and the symbolic planner over the shared symbolic action space. While the symbolic planner $\pi_{\text{tool}}$ is deterministic and produces a single action given $(o_t, g)$, we convert its output into a one-hot distribution by assigning a probability of 1.0 to the selected action and 0.0 to all others. This one-hot (categorical) distribution is then used to compute the KL divergence against the LM policy's predicted distribution. We will clarify this formulation in the problem definition section of the revised version.
>
> - *W5*: We appreciate the reviewer's suggestion. To distinguish between the input and output forms of $M$, we will use subscripts $M_\mathrm{in}$ and $M_\mathrm{out}$ in the revised version. This distinction will also be reflected in the symbol table for clarity.
>
> [1] Daya Guo et al. "Deepseek-r1: Incentivizing reasoning capability in llms via reinforcement learning". arXiv preprint 2025.
>
> [2] Malte Helmert. "The fast downward planning system”. Journal of Artificial Intelligence Research 2006.
>
> [3] Ling Yang et al. "Buffer of thoughts: Thought-augmented reasoning with large language models". Advances in Neural Information Processing Systems 2024.
>
> ### Method: ###
>
> - *W1*: Through the review process, we recognized the importance of making this point explicit. In the revised version, we will clarify that Eq. (1) specifies the overall objective of maximizing task success while minimizing KL divergence from the symbolic planner, and that Eq. (9) is the practical loss function used during optimization. This loss approximates Eq. (1) via behavior cloning and includes an additional VQ loss to train the procedure-book.
>
> ### Results: ###
>
> - *W1*: Thank you for the helpful suggestion. We agree that a side-by-side example comparing successful and failed plans can better illustrate how NeSyPr's contrastive planning mechanism contributes to improved performance. We plan to include qualitative examples that highlight how NeSyPr suppresses failure procedures, particularly in dynamically changing tasks. This will help clarify how procedural memory guides adaptive reasoning beyond what is captured by aggregate metrics alone.
>
> ---
>
> Lastly, your feedback has greatly helped us improve the clarity of our work and has given us an opportunity to learn and grow through this process as authors. We truly appreciate your time and thoughtful engagement.

---

### Note · Authors · 2025-08-11

Dear AC and Reviewers,

We sincerely thank all reviewers for their constructive feedback and Area Chair 4mLt for facilitating a meaningful discussion.

Thanks to the active engagement of Reviewer `sgZ5` and `cUia`, we addressed their points, particularly regarding:
- Implementation details of NeSyPr
- Formatting of procedural knowledge
- Principles enabling adaptive reasoning

Based on their feedback, in the revised version, we plan to:
- Add simple explanations for Eq. (1) and Eq. (9)
- Include a symbol table and qualitative examples
- Relocate method and baseline explanations from the Appendix

We also valued the feedback from Reviewer `La85` and `fttA`, and we have thoroughly addressed their concerns through clarifications and additional experiments.

Regarding Reviewer `La85`:
- Latency differences: We clarified that output token length dominates inference time in autoregressive models. We conducted a FLOPs-based analysis and added device-level profiling, showing NeSyPr achieves the best trade-off in success rate, latency, and compute cost.
- Role of pretrained representations: We conducted ablation (scratch, full finetune, LoRA finetune; with/without NeSyPr), showing NeSyPr consistently improves performance, with pretrained weights further enhancing proceduralization.
- Catastrophic forgetting: We explained that NeSyPr uses LoRA with a frozen backbone, verified that pretrained capabilities are preserved, and demonstrated continual learning without forgetting.

Regarding Reviewer `fttA`:
- Cross-environment generalization: We clarified that symbolic format alignment (PDDLGym vs. ALFWorld) is non-trivial and left for future work, but added experiments across PDDLGym domains showing partial generalization and potential.
- Prompt-based baseline: We conducted additional experiments with LLM-Planner (GPT-4.1), showing NeSyPr achieves comparable or better performance at much lower inference cost and without reliance on online API access.
- Ablation study: We referred to Table 6 with variant results, confirming that contrastive planning improves robustness.
- Interpretability: We explained that procedural vectors encode latent condition-action patterns, goal schemas, and operator structures. Although not symbolically interpretable, behavioral analysis reveals that similar tasks activate similar procedure-units, indicating a meaningful structure.

We hope our work and the review discussions are well reflected in the final evaluation.

Sincerely,

The Authors

---

### Decision · Program_Chairs · 2025-09-17

**Decision:**

Accept (poster)

**Comment:**

This paper presents NeSyPr, a neurosymbolic framework that compiles symbolic planning knowledge into procedural memory for efficient embodied reasoning without online symbolic planners. While reviewers initially raised concerns about clarity, computational claims, and generalization, the authors provided comprehensive rebuttals with additional experiments demonstrating genuine efficiency gains, cross-domain transfer capabilities and thorough baseline comparisons. Despite limitations in cross-environment generalization and procedural memory interpretability, the work addresses an important practical problem with a novel approach grounded in cognitive architecture theory, supported by comprehensive evaluation across multiple benchmarks. The substantial efficiency improvements for resource-constrained embodied systems, combined with the authors' thorough responses leading reviewers to favorable score updates, justify acceptance of this technically solid contribution to an emerging research direction.